# Functional data analysis for multivariate distributions through Wasserstein slicing

**Han Chen**
Department of Statistics
University of California, Davis
Davis, CA 95616
nahchen@ucdavis.edu

**Hans-Georg Müller**
Department of Statistics
University of California, Davis
Davis, CA 95616
hgmueller@ucdavis.edu

## Abstract

The modeling of samples of distributions is a major challenge since distributions do not form a vector space. While various approaches exist for univariate distributions, including transformations to a Hilbert space, far less is known about the multivariate case. We utilize a transformation approach to map multivariate distributions to a Hilbert space via a Wasserstein slicing method that is invertible. This approach combines functional data analysis tools, such as functional principal component analysis and modes of variation, with the facility to map back to interpretable distributions. We also provide convergence guarantees for the Hilbert space representations under a broad class of such transforms. The method is illustrated using joint systolic and diastolic blood pressure data.

## 1 Introduction

Data that consist of multivariate distributions are becoming increasingly available. Instances of such data encompass anthropometric data distributions [24], stock price data distributions [19] and distributions of temperatures in climate modeling. Analyzing distributional data presents distinctive challenges, as they do not form a vector space, rendering conventional methods designed for vector and functional data inapplicable.

For data that correspond to samples of univariate distributions, invertible global transformations such as the log quantile density transform and log hazard transform have been employed to map distributions to a Hilbert space $L^2$ [35]. While various maps were proposed to embed metric spaces into a Hilbert space, starting with Schoenberg's kernel embedding [43, 44] and also including the centered log-ratio transformation [23, 30, 24], these maps are generally not 1:1 and therefore not invertible. However, invertibility is crucial as functional principal component analysis or regression in Hilbert space involves elements that can be located anywhere in the Hilbert space. To get back to the original distribution space, which is a must for interpretability and visualization, therefore requires invertibility of the map.

Another example of such a non-invertible embedding for the Wasserstein space of univariate distributions is to use Riemannian type log maps to map densities to tangent bundles. The resulting tangent spaces, are Hilbert spaces, where the development of principal component analysis [5] and regression models [12] is straightforward, however the reverse exp map is not defined on the entire Hilbert space but only on a convex cone and this poses a major problem, as only ad hoc solutions such as projecting elements outside the invertibility cone on the cone are available. Statistical regression models targeting univariate distributions as responses and Euclidean predictors can be implemented with Fréchet regression [36], where also other distribution-to-distribution regression models with viable implementations have been proposed [16, 49].

39th Conference on Neural Information Processing Systems (NeurIPS 2025).

While these approaches have been developed for univariate distributions, multivariate distributional data have been less investigated; while kernel embeddings are still available, the invertibility problem remains. For regression with multivariate distributions, [34] introduce a regression model but its scope is limited to the case of multivariate Gaussian distributions, while [17] provides some theoretical results for a more general settings for an optimal transport map regression approach in analogy to the case of univariate distributions [16]. Another approach utilizes a Bayes Hilbert space over bivariate domains to model a time series of copula probability density functions [19], further developed for bivariate distributional data through spline representations for bivariate densities in the Bayes Hilbert space [24]. The Bayes Hilbert space proposals do not offer theoretical support and also do not address the estimation of densities from available random samples that they generate. Wasserstein geodesic principal component analysis represents variability among probability measures along geodesics in Wasserstein space, offering closed-form solutions for the case of Gaussian distributions and neural-network approximations for more general scenarios. However, it lacks an explicit inverse map and does not guarantee invertibility for non-Gaussian distributions [45].

Our goal in this paper is to extend the univariate log quantile density and log hazard transformation method [35] to the case of multivariate distributions. Then functional data analysis methodology, including functional principal component analysis (FPCA), functional modes of variation and functional regression can be applied in the Hilbert space into which the distributions are mapped. The densities corresponding to the Hilbert space representations are then obtained through a regularized inverse map [9]. Theoretical results regarding the convergence of these representations in density space are derived under regularity conditions, drawing strength from known results for FPCA and addressing the complexities introduced by the transformations. Given that multivariate distributions are often not fully observed and need to be recovered from random samples generated by the underlying distributions, we also consider the challenges introduced by the estimation step.

## 2 Preliminaries

### 2.1 Univariate Density Transformation

Let $\mathcal{G} = \{g \in \mathcal{C}^0(\mathcal{Z}) : g > 0, \int_0^1 g(z)dz = 1\}$ be the space of strictly positive densities on a common compact interval $\mathcal{Z}$. Without loss of generality, assume the support for the univariate density function is $\mathcal{Z} = [0, 1]$. [35] propose a family of transformations $\psi : \mathcal{G} \to L^2(\mathcal{S})$ that satisfy assumptions (L1)-(L3) that are listed in Appendix A. These transformations map continuous univariate densities to a new space $L^2(\mathcal{S})$, where $\mathcal{S}$ is a compact interval.

Here we focus on the log quantile density (LQD) transformation, while the log hazard transformation is another prominent example for this class of transformations and is reviewed in Appendix D. Choosing $\mathcal{S} = [0, 1]$, the LQD transformation $\psi_Q : \mathcal{G} \to L^2(\mathcal{S})$ is given by

$$\psi_Q(g)(s) = -\log\left(g(G^{-1}(s)\right), \quad s \in \mathcal{S}, \tag{1}$$

where $G(z) = \int_0^z g(u)du$ is the cumulative distribution function and $G^{-1}$ is the corresponding quantile function supported on $\mathcal{S}$. The inverse of a continuous function $\varphi$ on $\mathcal{S}$ can be written as $\exp\{-\varphi(G_\varphi(z))\}$, where $G_\varphi^{-1}(s) = \int_0^s e^{\varphi(u)}du$. Since $G_\varphi^{-1}(1)$ is not fixed, a boundary adjustment guarantees that the inverse will be a density on $\mathcal{Z} = [0, 1]$ [37],

$$\psi_Q^{-1}(\varphi)(z) = \alpha_\varphi \exp\{-\varphi\left(G_\varphi(z)\right)\}, \quad G_\varphi^{-1}(s) = \alpha_\varphi^{-1} \int_0^s e^{\varphi(u)}du,$$

where $\alpha_\varphi = \int_0^1 e^{\varphi(u)}du$. The LQD transformation is an element of a family of a larger class of univariate density transformations; however, similar transformations are currently not available for multivariate density functions. We fill this gap by introducing a generalized multivariate transformation in Section 3.2, which will be illustrated with examples based on multivariate extensions of the LQD transformation.

### 2.2 Density Slicing Transformation

To connect multivariate and univariate distributions, we use the Radon transform $\mathcal{R}$ [40], which is an integral transform that takes an integrable $p$-dimensional function and maps it to an infinite set of

integrals over hyperplanes in $\mathbb{R}^p$. Let $\theta$ be a unit vector in $\Theta = \{z \in \mathbb{R}^p, \|z\|_2 = 1\}$ and $z$ an element in $\mathbb{R}$. We denote affine hyperplanes by $l_{z,\theta} = \{x \in \mathbb{R}^p : \langle x, \theta \rangle = z\}$ and select an orthonormal basis $\{\theta, e_1, ..., e_{p-1}\}$ in $\mathbb{R}^p$ such that $\langle \theta, e_j \rangle = 0$ and $\langle e_j, e_l \rangle = \delta_{jl}$, for $j, l = 1, ..., p-1$. We then define the $p$-dimensional Radon transform $\mathcal{R}$ through the integral over $l_{z,\theta}$ as follows,

$$\mathcal{R}(f)(\theta, z) = \int_{\mathbb{R}^{p-1}} f\left(z\theta + \sum_{j=1}^{p-1} s_j e_j\right) ds_1 \cdots ds_{p-1}, \quad \text{for } \theta \in \Theta \text{ and } z \in \mathbb{R}, \qquad (2)$$

where $f$ is a $p$-dimensional multivariate density function supported on $D$.

The inverse Radon transform has undergone thorough investigation, both theoretically and numerically [1, 22, 31]. The commonly used method to reconstruct the original multivariate distribution from its Radon transform $\phi$ is through the filtered back-projection given by

$$\mathcal{R}^{-1}(\phi)(x) = \frac{1}{2(2\pi)^p} \int_{\Theta} \int_{\mathbb{R}} \mathcal{J}(\phi(\theta, \cdot))(v) e^{ir\langle \theta, x \rangle} |v|^{p-1} dv d\theta, \qquad (3)$$

where the Fourier transform is $\mathcal{J}(g)(v) = \int_{\mathbb{R}} g(u) e^{-iuv} du$ for all $v \in \mathbb{R}$. There are cases where Radon transforms are bona fide probability densities, while the inverse is not, such as the Wigner quasi-probability distribution in quantum state tomography [2]. Moreover, the inverse map $\mathcal{R}^{-1}$ is not continuous, and small reconstruction errors in $\mathcal{R}(f)$ can be magnified [15]. To address this, a regularized inverse [15, 25, 9] using a filter function in the Fourier domain is commonly applied,

$$\check{\mathcal{R}}_\tau^{-1}(\phi)(x) = \frac{1}{2(2\pi)^p} \int_{\Theta} \int_{\mathbb{R}} \mathcal{J}(\phi(\theta, \cdot))(v) e^{ir\langle \theta, x \rangle} |v|^{p-1} \mathbf{1}_{|v| \leq \tau} dv d\theta. \qquad (4)$$

Note that the regularization parameter $\tau$ balances approximation accuracy and the continuity of the inverse map [9]. As this regularized inverse is not guaranteed to be a multivariate density function, normalization is applied to map the resulting $L^1$ function into the multivariate density space $\mathcal{F}$ via

$$\mathcal{R}_\tau^{-1}(\phi)(x) = \begin{cases} \left|\check{\mathcal{R}}_\tau^{-1}(\phi)(x)\right| / \int_D \left|\check{\mathcal{R}}_\tau^{-1}(\phi)(x)\right| dz & \text{if } \int_D \left|\check{\mathcal{R}}_\tau^{-1}(\phi)(x)\right| dx > 0, \\ 1/|D| & \text{otherwise.} \end{cases} \qquad (5)$$

Here $|D|$ represents the Lebesgue measure of $D$ which is the domain of the multivariate density function $f$. Note that the Radon transform and its inverse are part of a generalized slicing transform family, for which an explicit invertible embedding between multivariate and sliced univariate densities was introduced in [9], where further details can be found.

# 3 Multivariate Density Transformation

## 3.1 Assumptions for Multivariate Densities

By convention, $\mathcal{C}^0$ denotes the set encompassing all continuous functions while $\mathcal{C}^1$ denotes the set consisting of all continuously differentiable functions. Let $\| \cdot \|_2$ be the $L^2$ metric and $\| \cdot \|_\infty$ be the uniform metric for measurable functions. Throughout, $C_0, C_1, \ldots$ represent various constants, and their dependency on pertinent quantities $\varrho$ will be denoted by writing $C_0(\varrho), C_1(\varrho), \ldots$.

We assume that the multivariate distributions that we consider have density functions and that their common support is $D \subset \mathbb{R}^p$, where

(D1) The support set $D \subset \mathbb{R}^p$ is compact and convex.

Denote the space of strictly positive multivariate density functions on $\mathbb{R}^p$ with compact support $D$ by

$$\mathcal{F} = \left\{ f(x) \in L^1(\mathbb{R}^p) : f(x) > 0, \int_{\mathbb{R}^p} f(x) dx = 1, \text{support}(f) = D, f \text{ satisfies (F1)} \right\}.$$

(F1) For all $f \in \mathcal{F}$, there exists a constant $M_0 > 0$ such that $\max\{\|f\|_\infty, \|1/f\|_\infty\} \leq M_0$ and for a $\kappa \geq p + 1$, $f$ is continuously differentiable of order $\kappa$ on $D \subset \mathbb{R}^p$ and has uniformly bounded partial derivatives.

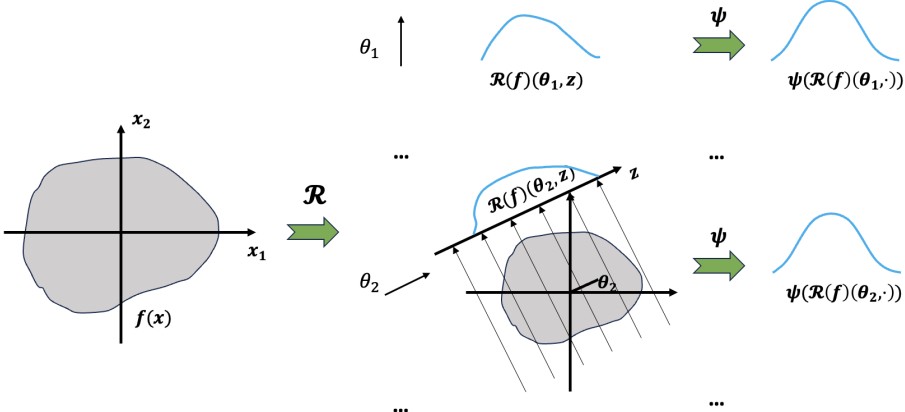

Figure 1: Scheme for the Radon Log Quantile Density (RLQD) transformation for the bivariate case: Slicing the bivariate density function along each projection (represented by an angle) through the Radon transform, followed by the slice-wise log quantile density transform.

## 3.2 Multivariate Density Transformation

We introduce multivariate density transformations $\Psi : \mathcal{F} \to L^2(\Theta \times \mathcal{S})$ from multivariate density functions to a Hilbert space as follows. Let $\psi, \psi^{-1}$ be the generalized univariate density transformation and its inverse reviewed in Section 2.1, then

$$\Psi(f)(\theta, s) = \psi\left(\mathcal{R}(f)(\theta, \cdot)\right)(s), \quad \text{for } (\theta, s) \in \Theta \times \mathcal{S}, \ f \in \mathcal{F}. \tag{6}$$

By using the regularized inverse of the Radon transform, the regularized inverse can be defined for any continuous function $\zeta$ as

$$\Psi_\tau^{-1}(\zeta)(x) = \mathcal{R}_\tau^{-1}\left(\left(\psi^{-1}(\zeta(\theta, \cdot))\,(z)\right)(x). \tag{7}$$

A concrete example is the Radon log quantile density (RLQD) transformation $\Psi_Q$,

$$\Psi_Q(f)(\theta, s) = \psi_Q\left(\mathcal{R}(f)(\theta, \cdot)\right)(s) = -\log\left(\mathcal{R}(f)(\theta, R_\theta^{-1}(s)\right), \text{ for } (\theta, s) \in \Theta \times \mathcal{S}, \ f \in \mathcal{F}, \tag{8}$$

where $R_\theta(z) = \int_0^z \mathcal{R}(f)(\theta, u)du$ is the cumulative distribution function along $z$ when fixing the first argument $\theta$ and $R_\theta^{-1}(s)$ is the corresponding sliced quantile function supported on $[0, 1]$. The inverse map can be defined for any continuous function $\zeta$ as

$$\Psi_{Q,\tau}^{-1}(\zeta)(x) = \mathcal{R}_\tau^{-1}\left(\psi_Q^{-1}\left(\zeta(\theta, \cdot)\right)(z)\right)(x). \tag{9}$$

The steps defining the RLQD transformation are visualized in Fig. 1.

For the theoretical analysis of these transformations we require certain smoothness and regularity conditions. We show that when the univariate density transformation $\psi$ satisfies assumptions (L1)-(L3) in the Appendix , the corresponding multivariate density transformation $\Psi$ according to (6) satisfies the four smoothness properties (T1)-(T4) in Appendix A.

## 4 Representations for Transformed Distributions

### 4.1 Metric for Multivariate Distributions

Let $\mathcal{B}(D)$ be the Borel $\sigma$-algebra on $D$. We focus on random probability distributions on $(D, \mathcal{B}(D))$ and each probability distribution is characterized by a density function with respect to Lebesgue measure. While classical metrics such as $L^1$, $L^2$, $L^\infty$ or total variation metric [14] are commonly used, recent work has emphasized the Fisher–Rao [13, 50] and Wasserstein metrics [18, 32, 26] for samples of distributional data. The 2-Wasserstein distance between two densities is defined as

$$d_W^2(f_1, f_2) = \inf_{\substack{\mu \in \Pi(f_1, f_2), \\ (Z_1, Z_2) \sim \mu}} E(\|Z_1 - Z_2\|^2), \quad \text{for all } f_1, f_2 \in \mathcal{F}.$$

Here $Z_1$ and $Z_2$ are random variables on $\mathbb{R}^p$, and $\Pi(f_1, f_2)$ is the space of joint probability measures on $D \times D$ with marginal densities $f_1$ and $f_2$. For one-dimensional distributions, the 2-Wasserstein metric corresponds to the $L^2$ distance between their quantile functions,

$$d_W^2(g_1, g_2) = \int_0^1 \left( G_1^{-1}(s) - G_2^{-1}(s) \right)^2 ds, \quad \text{for all } g_1, g_2 \in \mathcal{G},$$

where $G_1^{-1}, G_2^{-1}$ are the corresponding quantile functions of $g_1, g_2$. Obtaining the Wassestein distance for the multivariate distribution involves intricate algorithms and does not have an analytic solution [39, 38]. An alternative approach is the sliced Wasserstein distance [7]. It offers a computationally more efficient solution compared to the conventional Wasserstein distance and is defined as

$$d_{SW}^2(f_1, f_2) = \int_{\theta \in \Theta} d_W^2 \left( \mathcal{R}(f_1)(\theta, \cdot), \mathcal{R}(f_2)(\theta, \cdot) \right) d\theta, \quad \text{for all } f_1, f_2 \in \mathcal{F}, \tag{10}$$

where the Radon transform $\mathcal{R}$ is introduced in (2).

## 4.2 Transformed Distribution Representations

Suppose the random densities are generated as $f \sim \mathfrak{F}$ where $\mathfrak{F}$ is a probability measure on $\mathcal{F}$. Applying the transformation in Section 3 yields a random element in the Hilbert space $X := \Psi(f) \in L^2(\Theta \times \mathcal{S})$, which corresponds to the realization of a stochastic process. This enables functional data analysis [47] and linear representations in the Hilbert space $L^2$, with results mapped back to the density space via the regularized inverse $\Psi_\tau^{-1}$ that was introduced in (9).

Denote the mean function of the process $X$ by $\nu(\theta, s) = E(X(\theta, s))$ and the covariance function by $H(\theta_1, s_1, \theta_2, s_2) = \text{Cov}(X(\theta_1, s_1), X(\theta_2, s_2))$, where the eigenfunctions of the covariance operator associated with the kernel $H$ are denoted by $\{\gamma_k(\theta, s)\}_{k=1}^\infty$ and their corresponding eigenvalues by $\{\lambda_k\}_{k=1}^\infty$. The transformed process can be represented through the Karhunen-Loève expansion [27]

$$X(\theta, s) = \nu(\theta, s) + \sum_{k=1}^\infty \chi_k \gamma_k(\theta, s), \quad (\theta, s) \in \Theta \times \mathcal{S}, \tag{11}$$

with the principal components $\chi_k = \int_{\Theta \times \mathcal{S}} (X(\theta, s) - \nu(\theta, s)) \gamma_k(\theta, s) d\theta ds$. This expansion exhibits an optimality characteristic where the initial $K$ terms constitute the $K$-dimensional representation of $X(s, t)$ that minimizes the unexplained variance. Empirical estimators of $\hat{\nu}, \hat{H}, \hat{\gamma}_k, \hat{\lambda}_k, \hat{\chi}_k$ can be obtained based on the sample of transformed processes [4, 20, 3, 47]. For extensions to multivariate functional data, including product and marginal FPCA, see [11, 10].

Truncating the Karhunen-Loève expansion to the first $K$ components yields an $L^2$-optimal approximation; with judicious choice of $K$ it captures a pre-specified fraction of the variation,

$$X_K(\theta, s) = \nu(\theta, s) + \sum_{k=1}^K \chi_k \gamma_k(\theta, s), \quad (\theta, s) \in \Theta \times \mathcal{S}. \tag{12}$$

To interpret the contribution of each principal component in the original density space, we extend the notion of univariate transformation modes of variation [35] to the multivariate setting. Specifically, we compute modes of variation in the transformed $L^2$ space and map them back to the density space via the regularized inverse. The $k$-th transformation mode of variation is defined as a family of functions indexed by $\alpha \in \mathbb{R}$,

$$m_k(x, \alpha, \tau) = \Psi_\tau^{-1} \left( \nu + \alpha \sqrt{\lambda_k} \gamma_k \right)(x). \tag{13}$$

The data application in Section 8 demonstrates that transformation modes of variation play an important role in understanding the effect of each principal component.

## 5 Estimation

In practice, random densities $\{f_i\}_{i=1}^n$ are often not fully observed and one has only have observations generated by each of these densities. In this more realistic case, a preliminary density estimation step

is needed, which is described in Appendix F. If the random densities $f_i$ are estimated by a kernel density estimator $\hat{f}_i$ through (31), the corresponding sample mean and variance are based on the new transformed sample $\{\check{X}_i\}_{i=1}^n = \{\Psi(\hat{f}_i)\}_{i=1}^n$,

$$\hat{\nu}(\theta, s) = \frac{1}{n}\sum_{i=1}^n \check{X}_i(\theta, s), \quad \hat{H}(\theta_1, s_1, \theta_2, s_2) = \frac{1}{n}\sum_{i=1}^n \check{X}_i(\theta_1, s_1)\check{X}_i(\theta_1, s_2) - \hat{\nu}(\theta_1, s_1)\hat{\nu}(\theta_2, s_2).$$

(14)

Another application of Mercer's theorem implies

$$\hat{H}(\theta_1, s_1, \theta_2, s_2) = \sum_{k=1}^\infty \hat{\lambda}_k \hat{\gamma}_k(\theta_1, s_1)\hat{\gamma}_k(\theta_2, s_2),$$

(15)

where $\hat{\lambda}_k$ and $\hat{\gamma}_k$ are estimated eigenvalues and eigenfunctions of the covariance operator associated with the covariance function $\hat{H}$. The principal components $\chi_{ik}$ are estimated as

$$\hat{\chi}_{ik} = \int_\Theta \int_{\mathcal{S}} \left(\check{X}_i(\theta, s) - \hat{\nu}(\theta, s)\right)\hat{\gamma}_k(\theta, s)d\theta ds$$

(16)

and the plugin estimator for the transformation modes of variation (13) is

$$\hat{m}_k(x, \alpha, \tau) = \Psi_\tau^{-1}\left(\hat{\nu} + \alpha\sqrt{\hat{\lambda}_k}\hat{\gamma}_k\right)(x).$$

(17)

Selection of tuning parameter $K$ and $\tau$ are further discussed in Appendix E.

# 6 Asymptotic Convergence

In this section we establish asymptotic properties of the empirical estimators proposed in Section 5. A key feature in the convergence rate is the spacing between eigenvalues, where we define

$$\sigma_k = \min_{1 \le j \le k}(\lambda_j - \lambda_{j+1}).$$

(18)

for any integer $k$. For the $i^{th}$ distribution, we assume there are $N_i$ independent observations available to estimate the density function $\hat{f}_i$ in (31), with details in the Appendix. The lower bound $N(n) = \min_{1 \le i \le n} N_i$ is relevant for the overall asymptotic convergence rate. For the following result, we require that the sample size per distribution is large enough so that the effect of density estimation can be neglected when deriving the asymptotic rates. Specifically, we require

(E1) $\lim_{n\to\infty} N(n)/n^r \ge c$ for some $r > 1 + p/4$, where $c > 0$ is a constant and $p$ denotes the dimension of the domain of the densities.

Convergence analysis of the representations is presented in Appendix G. For the following, we require an additional assumption (S1) which imposes restrictions on the eigencomponents. It reflects typical assumptions of exponentially or polynomially declining eigenvalues.

(S1) The sequence $K = K(n)$ of approximating $K$-representations is such that $\sigma_K^{-1}\lambda_K^{-1}n^{-1/2} = O(1)$. With $C_4$ as in (T5) in Appendix A, we use the following bound to derive the convergence of transformation modes of variation, where for a fixed constant $\alpha_0 > 0$,

$$S(\tau, K) = \max_{1 \le k \le K} \sup_{|\alpha| \le \alpha_0} C_4\left(\tau, \|X_{k,\alpha}\|_\infty, \|\partial X_{k,\alpha}(\theta, s)/\partial s\|_\infty, d_\infty\left(X_{k,\alpha}, \hat{X}_{k,\alpha}\right)\right),$$

(19)

$$= O\left(\tau^p \max_{1 \le k \le K}\left\{e^{6\|\gamma_k\|_\infty}\|\partial\gamma_k(\theta, s)/\partial s\|_\infty e^{2d_\infty\left(X_{k,\alpha}, \hat{X}_{k,\alpha}\right)}\right\}\right),$$

with $X_{k,\alpha} = \nu + \alpha\sqrt{\lambda_k}\gamma_k$ and $\hat{X}_{k,\alpha} = \hat{\nu} + \alpha\sqrt{\hat{\lambda}_k}\hat{\gamma}_k$. Under (S1), we have $d_\infty\left(X_{k,\alpha}, \hat{X}_{k,\alpha}\right) = O_p(1)$. When $K$ is fixed, $\|X_{k,\alpha}\|_\infty$ and $\|\partial X_{k,\alpha}(\theta, s)/\partial s\|_\infty$ can be uniformly bounded, implying $S(\tau, K) = O_p(\tau^p)$. When $K = K(n) \to \infty$, the convergence of $S(\tau, K)$ requires boundedness of $\|X_{k,\alpha}\|_\infty$ and $\|\partial X_{k,\alpha}(\theta, s)/\partial s\|_\infty$, which is implied by the boundedness of $\|\gamma_k\|_\infty$ and $\|\partial\gamma_k(\theta, s)/\partial s\|_\infty$. The following result provides the rate of convergence of the estimated transformation modes of variation in (17) to the true modes in (13), uniformly over the parameter $|\alpha| \le \alpha_0$ and for any constant $\alpha_0 > 0$.

**Theorem 1.** *Under assumptions (D1), (F1), (T1)-(T4), (E1), (K1)-(K2), for fixed $K$ and $\alpha_0$, $m_k(x, \alpha, \tau)$ and $\hat{m}_k(x, \alpha, \tau)$ as per* (13) *and* (17),

$$\max_{1 \leq k \leq K} \sup_{|\alpha| \leq \alpha_0} d_{SW}\left(m_k(x, \alpha, \tau), \hat{m}_k(x, \alpha, \tau)\right) = O_p(\tau^p n^{-1/2}).$$

*If we assume $K = K(n) \to \infty$ satisfies (S1), then*

$$\max_{1 \leq k \leq K(n)} \sup_{|\alpha| \leq \alpha_0} d_{SW}\left(m_k(x, \alpha, \tau), \hat{m}_k(x, \alpha, \tau)\right)$$
$$= O_p(\tau^p \max_{1 \leq k \leq K} \{e^{6\|\gamma_k\|_\infty}\|\partial\gamma_k(\theta, s)/\partial s\|_\infty\}\sigma_K^{-1} n^{-1/2}),$$

*where $\tau, \gamma_k, \sigma_K$ are as in equations* (4), (11) *and* (18).

**Corollary 1.** *When assuming exponential decay for the eigenvalues, i.e., $\lambda_k = ce^{-\theta k}$ for some constants $c, \theta > 0$, with $K(n) = \lfloor \frac{1}{4\theta}\log(n)\rfloor \to \infty$, Theorem 1 provides that*

$$\max_{1 \leq k \leq K(n)} \sup_{|\alpha| \leq \alpha_0} d_{SW}\left(m_k(x, \alpha, \tau), \hat{m}_k(x, \alpha, \tau)\right) =$$
$$O_p(\tau^p \max_{1 \leq k \leq K} \{e^{6\|\gamma_k\|_\infty}\|\partial\gamma_k(\theta, s)/\partial s\|_\infty\}n^{-1/4}).$$

We also present the convergence rate of the truncated density estimator in 25. See G for additional asymptotic convergence results in the Appendix.

**Theorem 2.** *Under assumptions (D1), (F1), (T1)-(T5), (E1), (K1)-(K2), assume exponentially decaying eigenvalues $\lambda_k = ce^{-\theta k}$ for some constants $c$ and $\theta$. With $K(n) = \lfloor \frac{1}{3\theta}\log(n)\rfloor \to \infty$,*

$$\max_{1 \leq i \leq n} d_{SW}\left(f_i(\cdot), \hat{f}_i(\cdot, K, \tau)\right) = O_p\left(\tau^{-(\kappa - p)} + \tau^p n^{-1/6}\right),$$

*and choosing $\tau \sim n^{1/6k}$ leads to*

$$\max_{1 \leq i \leq n} d_{SW}\left(f_i(\cdot), \hat{f}_i(\cdot, K, \tau)\right) = O_p\left(n^{-(\kappa - p)/6k}\right).$$

# 7  Experiments

We illustrate our methods by comparing the performance of three approaches: (1) Directly applying FPCA and modes of variation to estimated densities viewed as elements of $L^2$, ignoring density constraints [48]. The resulting functions are truncated to their positive part and renormalized to ensure valid densities; (2) Performing PCA in the tangent space (TPCA) of the Bures–Wasserstein manifold of covariance operators at a reference density and maps the resulting components back to the density space via the exponential map [29] and (3) the proposed RLQD method, which takes the constraints into account and where $K$-component approximations as well as modes of variation are guaranteed to reside in the density space and additional adjustments are not necessary,

We conducted Monte Carlo experiments under two simulation settings: (A) random bivariate normals and (B) mixtures of two bivariate normals. Details of the data generation process appear in Table 1. In both settings, we generate 50 random densities, each from latent parameters, and sample 200 observations per density. As evaluation criterion we use the fraction of variance explained $\hat{V}_K/\hat{V}_\infty$ as defined in Section E, where a higher fraction of variance explained for the same choice of $K$ reflects better representations. Boxplots depicting the fraction of variance explained with respect to the sliced Wasserstein distance for FPCA and RLQD approaches are in Figure 4. It is clear that in these settings the proposed RLQD method performs the best throughout in comparison with both the standard FPCA approach and the tangent FPCA approach. The advantage of RLQD becomes particularly evident beyond the first two modes, where FPCA's explained variance plateaus and TPCA yields diminishing returns. In contrast, RLQD continues to capture meaningful higher-order variation, demonstrating stronger structural representation and efficiency with fewer retained components.

We further visualize the modes of variation for the first two principal components under Setting A in Figures 2 and 3. Under the Gaussian assumption, both the proposed RLQD transformation and the classic FPCA exhibit similar patterns of variation. Specifically, the first mode primarily captures the overall intensity or concentration of the density around the mean, whereas the second mode reflects a rotational change in the orientation of the joint distribution.

Table 1: Simulation settings for bivariate normal and mixture distributions.

| Setting | Random components | Resulting density |
|---------|-------------------|-------------------|
| A | $\mu_i \sim \mathcal{U}[-0.2, 0.2]$ 
 $\rho_i \sim \mathcal{U}[0.4, 0.8]$ 
 $\log(s_{1i}) \sim \mathcal{U}[-0.3, 0.3]$ 
 $\log(s_{2i}) \sim \mathcal{U}[-0.3, 0.3]$ 
 $\Sigma_i = \begin{pmatrix} s_{1i}^2 & s_{1i}s_{2i}\rho_i \\ s_{1i}s_{2i}\rho_i & s_{2i}^2 \end{pmatrix}$ | $\{X_{ij}\}_{j=1}^{200} \sim \mathcal{N}(\mu_i, \Sigma_i)$ 
 truncated on $[-2, 2]^2$ 
 $i = 1, \ldots, 50$ |
| B | $\mu_{1,i}, \mu_{2,i} \sim \mathcal{U}[-0.2, 0.2]$ 
 $\rho_{1,i}, \rho_{2,i} \sim \mathcal{U}[0.4, 0.8]$ 
 $\log(s_{1,1i}), \log(s_{2,1i}) \sim \mathcal{U}[-0.3, 0.3]$ 
 $\log(s_{1,2i}), \log(s_{2,2i}) \sim \mathcal{U}[-0.3, 0.3]$ 
 $\Sigma_{1,i} = \begin{pmatrix} s_{1,1i}^2 & s_{1,1i}s_{1,2i}\rho_i \\ s_{1,1i}s_{1,2i}\rho_i & s_{1,2i}^2 \end{pmatrix}$ 
 $\Sigma_{2,i} = \begin{pmatrix} s_{2,1i}^2 & s_{2,1i}s_{2,2i}r_i \\ s_{2,1i}s_{2,2i}r_i & s_{2,2i}^2 \end{pmatrix}$ | $\{X_{ij}\}_{j=1}^{200} \sim \frac{1}{2}\mathcal{N}(\mu_{1,i}, \Sigma_{1,i})$ 
 $+ \frac{1}{2}\mathcal{N}(\mu_{2,i}, \Sigma_{2,i})$, 
 truncated on $[-2, 2]^2$ 
 $i = 1, \ldots, 50$ |

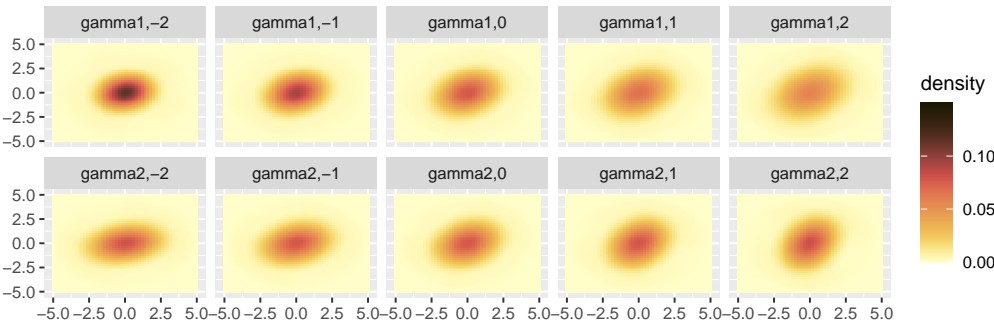

Figure 2: Proposed RLQD transformation modes of variation as per 13 for Setting A

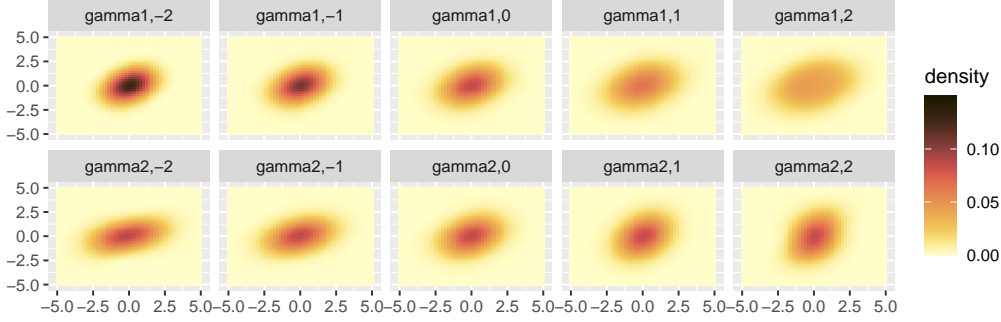

Figure 3: Classic FPCA Modes of variation for Setting A

# 8 Data Illustrations

We analyze blood pressure data from the Baltimore Longitudinal Study of Aging (BLSA) https://www.blsa.nih.gov/. The dataset comprises 16,715 bivariate measurements (SBP, DBP) for approximately 2,800 individuals, with the number of visits per individual ranging from 1 to 26 and ages spanning from 17 to 75. For each age group, we estimate joint two-dimensional densities using a kernel density estimator. The estimator is applied over 51 equidistant grid points in each direction,

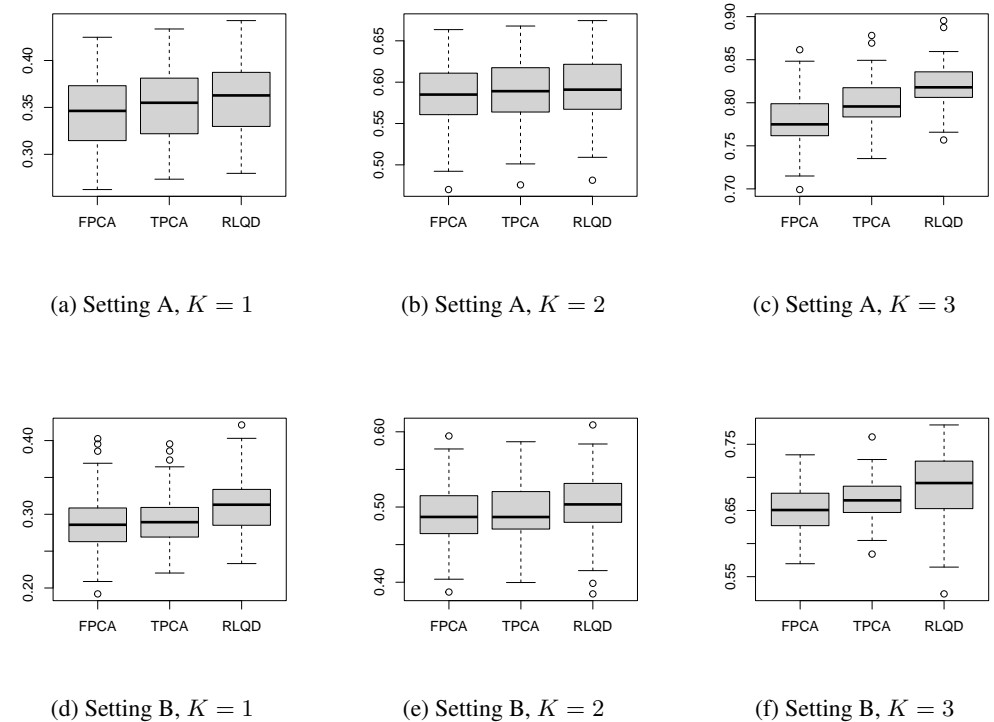

(a) Setting A, $K = 1$    (b) Setting A, $K = 2$    (c) Setting A, $K = 3$

(d) Setting B, $K = 1$    (e) Setting B, $K = 2$    (f) Setting B, $K = 3$

Figure 4: Boxplots show the fraction of variance explained (via sliced Wasserstein distance) across 100 simulations. Each row represents a different setting, with varying component numbers $K$. Larger values indicate that a higher fraction of variation is explained and are preferred. The proposed RLQD approach is compared with unconstrained FPCA and tangent PCA for densities.

covering the domains $[50, 205]$ for SBP and $[40, 125]$ for DBP with Gaussian kernel and bandwidths $h_{SBP} = 24$ and $h_{DBP} = 15$. The fitted densities (Figure 7 in the Appendix B) show a clear positive correlation between SBP and DBP, along with increasing variability at older ages, highlighting the need for low-dimensional representations that capture these distributional shifts.

Using the RLQD approach with $K = 2$ components, we achieve strong dimension reduction while explaining 87% of the variance in sliced Wasserstein distance as shown in Figure 8 of the Appendix. The corresponding transformation modes of variation in Figure 5 provide insight into principal patterns of change. The predominant mode of variation in the top row reflects a horizontal mean shift coupled with increasing variance, primarily in systolic blood pressure. This suggests that the primary source of variation is a shift in the mean and increased variance of systolic blood pressure. The second mode of variation highlights changes in the variation of diastolic blood pressure and in the covariance structure between diastolic and systolic blood pressures. The first two FPCA components explain 70% and 11%. Under a standard threshold of 85% cumulative variance, RLQD requires only 2 components for effective dimension reduction, whereas FPCA requires at least 3. The eigenfunctions and modes of variation produced by FPCA typically resemble Gaussian-shaped perturbations around the mean as presented in Figure 9 of the Appendix, which can be restrictive when modeling data with inherent skewness or non-Gaussian features. In contrast, RLQD captures more flexible, non-Gaussian modes that better reflect the underlying distributions. For example, the blood pressure data exhibit clear asymmetry, with a noticeable skew toward lower pressure levels. This is statistically confirmed by Mardia's test [28] for multivariate normality: both skewness (3063.39) and kurtosis (42.75) statistics are highly significant ($p$-values $< 0.001$), indicating a strong departure from multivariate normality. These findings demonstrate that RLQD not only provides more informative components with fewer dimensions but also produces representations that are more faithful to the structural properties of complex real-world distributions.

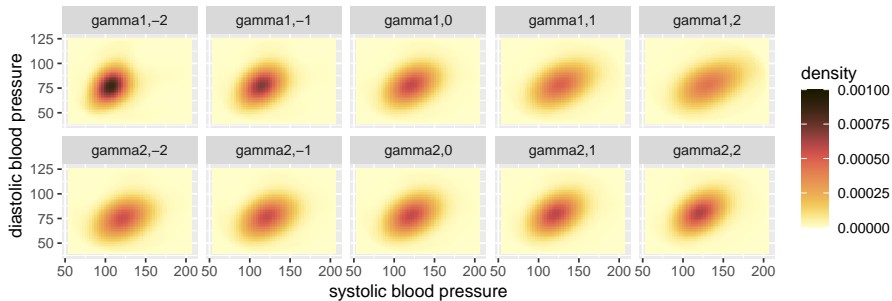

Figure 5: Transformation modes of variation as per (13) for bivariate blood pressure distributions. The first row illustrates the action of the first mode of variation $m_1$ and the second row that of the second mode of variation $m_2$ as the values of $\alpha$ are varied from $-2$ to $2$ as per (17).

One can also use the $K$-component representations—here with $K = 2$, yielding bivariate score vectors—to characterize evolving blood pressure patterns. Local linear smoothing of the scores against age is shown in Figure 6, using a Gaussian kernel with bandwidth 6, selected via leave-one-out cross-validation. The first score, $\chi_1$, associated with $\gamma_1$, shows an increasing trend with age, suggesting rising blood pressure variation and increased diastolic pressure in older individuals. The second score, $\chi_2$, linked to $\gamma_2$, increases until midlife then declines, indicating that diastolic pressure peaks in midlife. An additional case study on temperature data in the United States is provided in Appendix H.

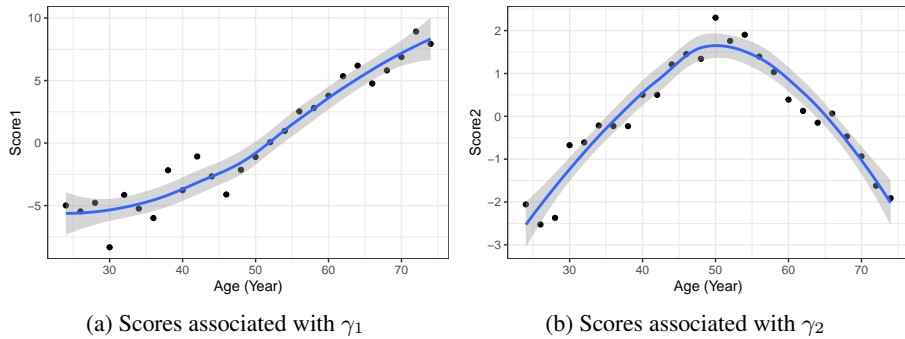

(a) Scores associated with $\gamma_1$        (b) Scores associated with $\gamma_2$

Figure 6: Blood pressure data score smoothing. Local linear smoothing of the score representations along age. Left: Smoothed first scores $\chi_1$ versus age. Right: Smoothed second scores $\chi_2$ against age.

## 9 Conclusion and Limitations

This work presents a transformation-based framework for the functional analysis of multivariate distributions. By combining Radon slicing with univariate log quantile density transforms, we map multivariate densities into a Hilbert space where classical FDA tools such as FPCA and modes of variation become applicable. The regularized inverse transform ensures that the structure of the original density space is preserved. While not the focus of this paper, we note that this framework facilitates distributional regression once vectorized representations are obtained. Limitations include the dependence on accurate density estimation from finite samples that is subject to the curse of dimensionality and the need for a regularization parameter.

## Acknowledgement and Disclosure of Funding

We would like to thank the reviewers for their constructive feedback. This research was partially supported by NSF grant DMS-2310450.

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

# A   Additional Assumptions

The univariate transformation $\psi$ is required to satisfy the following assumptions (L1)-(L3), where we denote the $L^2$ and uniform metrics as $d_2$ and $d_\infty$, respectively.

(L1) Let $g_1, g_2 \in \mathcal{G}$ with $g_1$ differentiable and $\|g_1'\|_\infty < \infty$. Set

$$\tilde{D}_0 \geq \max\left(\|g_1\|_\infty, \|1/g_1\|_\infty, \|g_2\|_\infty, \|1/g_2\|_\infty, \|g_1'\|_\infty\right).$$

There exists a const $\tilde{C}_0$ depending only on $\tilde{D}_0$ such that

$$d_2(\psi(g_1), \psi(g_2)) \leq \tilde{C}_0 d_2(g_1, g_2), \quad d_\infty(\psi(g_1), \psi(g_2)) \leq \tilde{C}_0 d_\infty(g_1, g_2).$$

(L2) Assume $g \in \mathcal{G}$ is differentiable with $\|g'\|_\infty < \infty$ and $\tilde{D}_1$ is a constant bounded below by $\max\left(\|g\|_\infty, \|1/g\|_\infty, \|g'\|_\infty\right)$. Then $\psi(g)$ is differentiable and there exists a constant $\tilde{C}_1 > 0$ depending only on $\tilde{D}_1$ such that

$$\|\psi(g)\|_\infty \leq \tilde{C}_1, \quad \|\psi(g)'\|_\infty \leq \tilde{C}_1.$$

(L3) Assume $\varphi_1, \varphi_2$ are continuous functions, where $\varphi_1$ is differentiable on $\mathcal{S}$ with $\|\varphi_2'\|_\infty < \infty$. There exist constants $c, \tilde{C}_2, \tilde{C}_3 > 0$ where $\tilde{C}_2\left(\|\varphi_1\|_\infty, \|\varphi_1'\|_\infty\right) \leq ce^{6\|\varphi_1\|_\infty}\|\partial\varphi_1(\theta, s)/\partial s\|_\infty$ and $\tilde{C}_3\left(d_\infty(\varphi_1, \varphi_2)\right) \leq ce^{2d_\infty(\varphi_1, \varphi_2)}$, such that

$$d_2\left(\psi^{-1}(\varphi_1), \psi^{-1}(\varphi_2)\right) \leq \tilde{C}_2 \tilde{C}_3 d_2(\varphi_1, \varphi_2),$$

where $\tilde{C}_2$ and $\tilde{C}_3$ are increasing in their respective arguments.

The multivariate transformation $\Psi$ is required to satisfy the following assumptions (T1)-(T5).

(T1) There exists a constant $C_1$ such that

$$d_2\left(\Psi(f_1), \Psi(f_2)\right) \leq C_1 d_2(f_1, f_2), \; d_\infty\left(\Psi(f_1), \Psi(f_2)\right) \leq C_1 d_\infty(f_1, f_2), \quad \text{for all } f_1, f_2 \in \mathcal{F}.$$

(T2) For all $f \in \mathcal{F}$, $\Psi(f)(\theta, s)$ is continuous and has a continuous partial derivative with respect to $s$ with a constant $C_2$ such that

$$\|\Psi(f)\|_\infty \leq C_2, \|\partial\Psi(f)(\theta, s)/\partial s\|_\infty \leq C_2, |\Psi(f)(\theta_1, s) - \Psi(f)(\theta_2, s)| \leq C_2\|\theta_1 - \theta_2\|_2.$$

(T3) An inverse transformation $\Psi^{-1}$ exists such that $\Psi^{-1} \circ \Psi(f) = f$ for all $f \in \mathcal{F}$.

(T4) There exists a sequence of approximating inverses $\Psi_\tau^{-1}, \tau \to \infty$, and a constant $C_3 = C_3(\tau) \leq c\tau^{-(\kappa-p)}$ for a constant $c$, such that

$$d_\infty\left(\Psi_\tau^{-1} \circ \Psi(f), f\right) \leq C_3, \quad \text{for all } f \in \mathcal{F},$$

where $C_3$ is decreasing to 0 as $\tau \to \infty$.

(T5) Let $\zeta_1, \zeta_2 : \Theta \times \mathcal{S} \to \mathbb{R}$ be continuous functions, where $\zeta_1$ has a continuous partial derivative $\partial\zeta_1(\theta, s)/\partial s$. There exist constants $c, C_4 > 0$ with

$$C_4 = C_4\left(\tau, \|\zeta_1\|_\infty, \|\partial\zeta_1(\theta, s)/\partial s\|_\infty, d_\infty(\zeta_1, \zeta_2)\right) \leq c\tau^p\|\partial\zeta_1(\theta, s)/\partial s\|_\infty e^{6\|\zeta_1\|_\infty}e^{2d_\infty(\zeta_1, \zeta_2)},$$

such that

$$d_\infty\left(\Psi_\tau^{-1}(\zeta_1), \Psi_\tau^{-1}(\zeta_2)\right) \leq C_4 d_2\left(\zeta_1, \zeta_2\right),$$

where $C_4$ is increasing in each of its four arguments.

Assumptions (T1) and (T2) ensure the continuity and the boundedness of the forward transformation, while assumption (T3) guarantees invertibility on the image space $\Psi(\mathcal{F})$. Note that $\Psi(\mathcal{F})$ is not guaranteed to cover the entire $\mathcal{C}^0$ space and the inverse transformation $\Psi^{-1}$ is only defined on the image space $\Psi(\mathcal{F})$. A sequence of approximating inverses $\Psi_\tau^{-1}$ is provided in assumption (T4) and (T5) and is used to map $L^2$ elements back to $\mathcal{F}$. Note that the regularization parameter $\tau$ controls the approximation accuracy and the roles of $\zeta_1$ and $\zeta_2$ in assumption (T5) could be reversed.

# B   Additional Figures

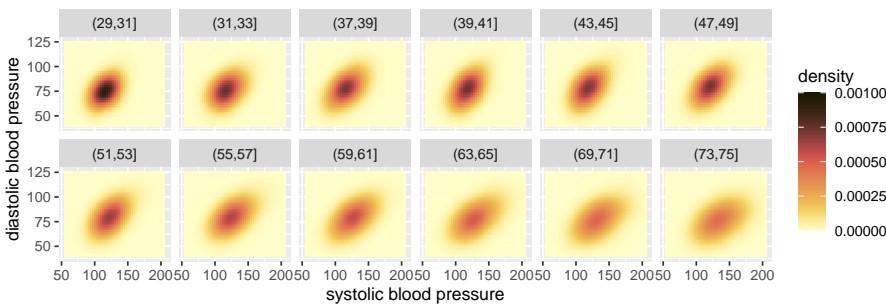

Figure 7: Fitted kernel density surfaces for bivariate blood pressure data for selected age groups.

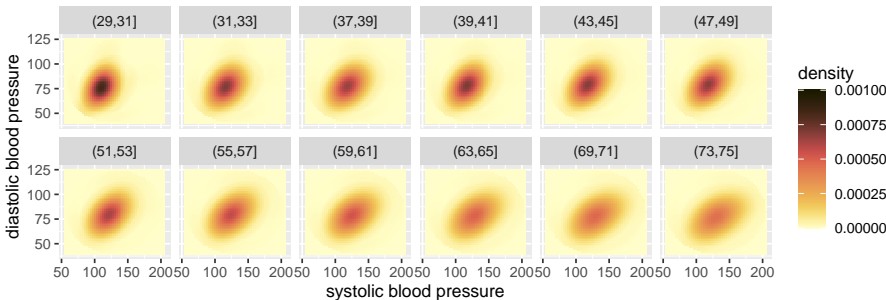

Figure 8: Reconstructed two-dimensional blood pressure distributions using the proposed RLQD approach, with sliced Wasserstein variance explained of 0.87—0.73 from the first component and 0.14 from the second. The two-dimensional representations capture most of the variation in the original density space and enable substantial dimension reduction.

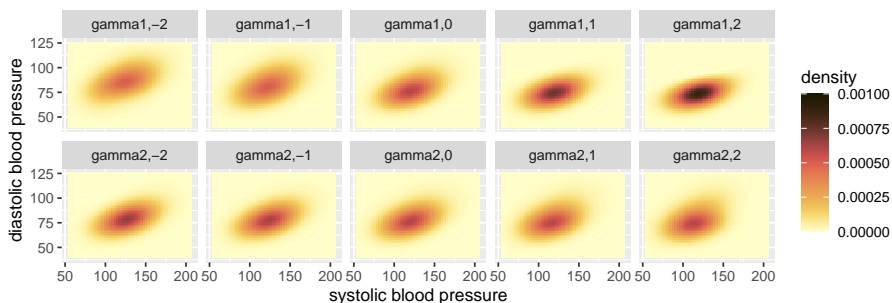

Figure 9: Modes of variation of the first two components by using standard FPCA approach

# C   Reconstruction of the L2 process

**Proposition 1.** *Assume (D1), (F1) and (T2)-(T5). Let $f \in \mathcal{F}$ and $\zeta$ be a continuous function on $L^2(\Theta \times \mathcal{S})$. Then*

$$d_{SW}\left(f, \Psi_\tau^{-1} \circ \zeta\right) = O\left(C_3 + C_4 d_2\left(\Psi(f), \zeta\right)\right), \quad as\ \tau \to \infty.$$

*Here, $C_3$, $C_4$, and $\Psi$ are defined as per assumptions (T4)-(T5) and in equation* (6).

This bound indicates that by approximating the transformed density within the transformed $L^2$ space, one can obtain a consistent reconstruction of the multivariate density through the regularized inverse map $\Psi_\tau^{-1}$ introduced in (9), where the rate of convergence can be further specified as in the Proposition.

## D   Radon Log Hazard and Log Quantile Density Transformations

We review here an additional specific univariate density transformation, the log hazard transformation [35]. We adopt the assumption that has been made in Section 2, namely that the support for a univariate density function is $\mathcal{Z} = [0, 1]$. As hazard functions diverge at the right endpoint 1 of the support of the distribution, we consider quotient spaces by equating densities that are equal on a subdomain $\mathcal{S} = [0, 1_\upsilon]$, where $1_\upsilon = 1 - \upsilon$ for some $0 < \upsilon < 1$. The log hazard transformation $\psi_H : \mathcal{G} \to L^2(\mathcal{S})$ is expressed as follows:

$$\psi_H(g)(s) = \log\left\{\frac{g(s)}{1 - G(s)}\right\}, \quad s \in \mathcal{S}, \tag{20}$$

where $G(s) = \int_0^s g(z)dz$ is the cumulative distribution function. The inverse of a continuous function $\varphi$ on $\mathcal{S}$ is

$$\psi_H^{-1}(\varphi)(z) = \begin{cases} \exp\left\{\varphi(z) - \int_0^z e^{\varphi(u)}du\right\} & \text{for } z \in [0, 1_\upsilon], \\ \upsilon^{-1}\exp\left\{-\int_0^{1_\upsilon} e^{\varphi(u)}du\right\} & \text{for } z \in [1_\upsilon, 1]. \end{cases}$$

Employing the Radon transform $\mathcal{R}$ as defined in (2), the Radon log hazard transformation (RLH) $\Psi_H$ is

$$\Psi_H(f)(\theta, s) = \psi_H(\mathcal{R}(f)(\theta, \cdot))(s) = \log\left\{\frac{\mathcal{R}(f)(\theta, s)}{1 - R_\theta(s)}\right\}, \tag{21}$$

where $R_\theta(s) = \int_{-\infty}^s \mathcal{R}(f)(\theta, u)du$ is the cumulative distribution function along $\mathcal{R}(f)(\theta, \cdot)$, fixing the first argument at $\theta$.

**Corollary 2.** *For $C_3, C_4$ as in assumption (T4)-(T5), when using the RLH transform $\Psi_H$ as per (21), it holds that $C_3(\tau) \sim \tau^{-(\kappa-p)}$ and $C_4\left(\tau, \|\zeta_1\|_\infty, \|\partial\zeta_1(\theta, s)/\partial s\|_\infty, d_\infty(\zeta_1, \zeta_2)\right) \sim \tau^p e^{2\|\zeta_1\|_\infty} e^{d_\infty(\zeta_1, \zeta_2)}$ as $\tau \to \infty$.*

Similarly, for Radon log quantile density (RLQD) transformation, we have

**Proposition 2.** *If (D1), (F1) and (L1)-(L3) hold, assumptions (T1)-(T5) are satisfied for the multivariate density transformation $\Psi$ in (6). When using the RLQD transformation $\Psi_Q$ as per (8), we have $C_3(\tau) \sim \tau^{-(\kappa-p)}$ and $C_4\left(\tau, \|\zeta_1\|_\infty, \|\partial\zeta_1(\theta, s)/\partial s\|_\infty, d_\infty(\zeta_1, \zeta_2)\right) \sim \tau^p \|\partial\zeta_1(\theta, s)/\partial s\|_\infty e^{6\|\zeta_1\|_\infty} e^{2d_\infty(\zeta_1, \zeta_2)}$.*

The following corollary follows directly from Corollary 2 and Theorem 1.

**Corollary 3.** *When assuming exponentially decaying eigenvalues, $\lambda_k = ce^{-\theta k}$ for some constants $c, \theta > 0$, with $K(n) = \lfloor \frac{1}{4\theta}\log(n)\rfloor \to \infty$, it holds that*

$$\max_{1 \le k \le K(n)} \sup_{|\alpha| \le \alpha_0} d_{SW}\left(m_k(x, \alpha, \tau), \hat{m}_k(x, \alpha, \tau)\right) = O_p\left(\tau^p \max_{1 \le k \le K}\left\{e^{2\|\gamma_k\|_\infty}\right\} n^{-1/4}\right).$$

*Furthermore, if the eigenfunctions correspond to the trigonometric basis, then*

$$\max_{1 \le k \le K(n)} \sup_{|\alpha| \le \alpha_0} d_{SW}\left(m_k(x, \alpha, \tau), \hat{m}_k(x, \alpha, \tau)\right) = O_p\left(\tau^p n^{-1/4}\right).$$

## E   Estimation and Selection of Tuning Parameter

We discuss the empirical estimation in Section 5. We now present the population version to provide a complete formulation and to facilitate the proofs.

Suppose we have a sample of independent random densities $\{f_i\}_{i=1}^n \sim \mathfrak{F}$. Writing $X_i = \Psi(f_i)$, empirical sample mean and variance are

$$\tilde{\nu}(\theta, s) = \frac{1}{n}\sum_{i=1}^n X_i(\theta, s), \quad \tilde{H}(\theta_1, s_1, \theta_2, s_2) = \frac{1}{n}\sum_{i=1}^n X_i(\theta_1, s_1)X_i(\theta_1, s_2) - \tilde{\nu}(\theta_1, s_1)\tilde{\nu}(\theta_2, s_2). \tag{22}$$

Mercer's theorem implies that the spectral decomposition of $\tilde{H}$ leads to

$$\tilde{H}(\theta_1, s_1, \theta_2, s_2) = \sum_{k=1}^\infty \tilde{\lambda}_k \tilde{\gamma}_k(\theta_1, s_1)\tilde{\gamma}_k(\theta_2, s_2), \tag{23}$$

where $\tilde{\lambda}_k$ and $\tilde{\gamma}_k$ are empirical eigenvalues and eigenfunctions of the covariance operator with the covariance kernel $\tilde{H}$.

In terms of the selection of tuning parameter, By selecting the first $K$ components in the expansion (11) and applying the regularized inverse transformation, the truncated representations can be obtained through

$$f_i(x, K, \tau) = \Psi_\tau^{-1}\left(\nu + \sum_{k=1}^K \chi_{ik}\gamma_k\right)(x), \tag{24}$$

with estimated version

$$\hat{f}_i(x, K, \tau) = \Psi_\tau^{-1}\left(\hat{\nu} + \sum_{k=1}^K \hat{\chi}_{ik}\hat{\gamma}_k\right)(x). \tag{25}$$

A simple and often well-working approach for the selection of the tuning parameter $K$ can be based on the fraction of variance explained in the density space. To implement this method, we need to first define the sliced Wasserstein Fréchet mean with respect to the metric $d_{SW}$, which is

$$f_\oplus = \operatorname*{arginf}_{q \in \mathcal{F}} E\left(d_{SW}(f, q)^2\right), \tag{26}$$

with empirical estimates $\hat{f}_\oplus = \operatorname{argmin}_{q \in \mathcal{F}} n^{-1}\sum_{i=1}^n d_{SW}^2(\hat{f}_i, q)$. We note that $f_\oplus$ and $\hat{f}_\oplus$ might be sets with more than one element. Further results regarding the sliced Wasserstein distance and sliced Wasserstein barycenter can be found in [38, 6, 7]. To quantify the variance explained in the density space, we use the Fréchet variance

$$V_\infty = E\left(d_{SW}(f, f_\oplus)^2\right), \tag{27}$$

which is unique irrespective of whether the Fréchet mean is unique or not. Its empirical estimate is $\hat{V}_\infty = \frac{1}{n}\sum_{i=1}^n d_{SW}(\hat{f}_i, \hat{f}_\oplus)^2$. The variance explained by the $K$ component representation is

$$V_K = V_\infty - E\left(d_{SW}(f, f_{K,\tau})^2\right), \tag{28}$$

where $f_{K,\tau} = \Psi_\tau^{-1}(\nu + \sum_{k=1}^K \chi_k \gamma_k)$ is the truncated representation of the density process. This variance explained is estimated by $\hat{V}_K = \hat{V}_\infty - \frac{1}{n}\sum_{i=1}^n d_{SW}\left(\hat{f}_i(\cdot), \hat{f}_i(\cdot, K, \tau)\right)^2$. Control of the tuning parameter $\tau$ is implemented by adjusting the sampling rate in the density domain, which determines the Nyquist frequency—the highest frequency representable without aliasing. This Nyquist frequency directly sets the cutoff for the regularized inverse Radon transform, thus controlling the level of smoothing in the reconstructed density [25, 41].

The fraction of variance explained is the ratio $V_K/V_\infty$, with estimated ratio $\hat{V}_K/\hat{V}_\infty$. The choice of $K$ is then the smallest $K$ that explains a fraction of variance above a given threshold $\vartheta$ with $0 < \vartheta < 1$, formally

$$K^* = \operatorname*{argmin}_K \left\{K : \frac{V_K}{V_\infty} > \vartheta\right\}, \tag{29}$$

with plug-in estimator

$$\hat{K}^* = \operatorname*{argmin}_K \left\{K : \frac{\hat{V}_K}{\hat{V}_\infty} > \vartheta\right\}. \tag{30}$$

Here the default choices of $\vartheta$ are $\vartheta = .95$ or $\vartheta = .9$.

## F    Random Distribution Estimation

We assume that the multivariate distributions we are dealing with are well-defined and have density functions on fixed domains, which might be unknown but encompass a fixed known domain $D$. The domain $D$ is as in assumption (D1) and is determined either by prior knowledge or selected by the user in practical scenarios. Our focus is on distributions truncated to the common domain $D$ and we presume that these truncated distributions are well-defined with density functions on $D$. Similar to the preliminary density estimation step as in Section S.3 of [9], we assume that there exists $\epsilon > 0$ such that $D_\epsilon = \bigcup_{z \in D} B(z, \epsilon)$ belongs to the domain of all underlying multivariate distributions, where $B(z, \epsilon)$ is a ball with radius $\epsilon$ centered at $z$. This assumption guarantees that boundary effects can be ignored in the density estimation step. The continuous differentiability assumption (F1) is extended to (F1$'$) below. Additional assumptions (P1), (K1), (K2) are also listed below.

(F1$'$) There exists a constant $M_0 > 0$ and an integer $\kappa \geq p + 1$ such that for all $f \in \mathcal{F}$, the density function $f$ is the density of a truncated distribution on the domain $D$, where the original distribution has a domain $D_f$ with $D_f \supset D_\epsilon$ and a density $f_{D_f}$ that is defined on $D_f$. It holds that $\max\{\|f_{D_f}\|_\infty, \|1/f_{D_f}\|_\infty\} \leq M_0$ on $D_\epsilon$ and that $f_{D_f}$ is continuously differentiable of order $k$ on $D_\epsilon$, with uniformly bounded partial derivatives. The target distributions with densities $f$ are the truncated versions of the $f_{D_f}$ on $D$, so that for all $z \in D$ one has $f(z) = f_{D_f}(z)/\int_D f_{D_f}(u)du$.

(P1) $N(n) = \min_{1 \leq i \leq n} N_i \geq cn^r$, where $N_i$ is the number of random observations for the $i$-th distribution, and the constants satisfy $r > 1 + p/4$ and $c > 0$.

(K1) The kernel function $\mathcal{K}$ as per (31) is a probability density function that has compact support and is a symmetric, bounded and $k$ times continuously differentiable function (without loss of generality, the support is assumed to be contained in the unit cube of $\mathbb{R}^p$).

(K2) For some $A > 0$ and $\omega > 0$, the class of functions $\mathcal{I}_b = \{\mathcal{K}\left(\frac{z - \cdot}{b}\right), z \in \mathbb{R}^p, b > 0\}$ satisfies

$$\sup_{\mathcal{P}} M\left(\mathcal{I}_b, L_2(\mathcal{P}), \varepsilon \|F\|_{L_2(\mathcal{P})}\right) \leq \left(\frac{A}{\varepsilon}\right)^\omega,$$

where $M(T, d, \varepsilon)$ denotes the $\varepsilon$-covering number of the metric space $(T, d)$, $F$ is the envelope function of $\mathcal{I}_b$ and the supremum is taken over the set of all probability measures on $\mathbb{R}^p$. The quantities $A$ and $\omega$ correspond to the Vapnik-Cervonenkis (VC) characteristic of $\mathcal{K}$.

Note that assumption (P1) ensures that the $L^2$ convergence of the multivariate density is faster than the parametric rate $n^{-1/2}$ while assumptions (K1)-(K2) are standard to derive the $L^2$ and uniform convergence of the multivariate density estimator. Let $\mu_{D_f}$ be a random probability distribution with density function $f_{D_f}$ that satisfies (F1'), from which random observations $W_1, ..., W_N$ are independently sampled. A standard kernel estimator $\check{f}$ is then applied to estimate $f_{D_f}$, and its truncated version $\hat{f}$ is employed to estimate the density $f$, which is the distribution $\mu_{D_f}$ truncated to the domain $D$.

$$\check{f}(x) = \frac{1}{Nh^p} \sum_{j=1}^{N} \mathcal{K}\left(\frac{x - W_j}{h}\right), \quad \hat{f}(x) = \check{f}(x) \bigg/ \int_D \check{f}(u)du, \quad x \in D \subset \mathbb{R}^p. \tag{31}$$

The following proposition is a direct consequence of Propositions S2 and S3 in [9].

**Proposition 3.** *Assume (D1), (F1$'$) and (K1). Choosing $h \sim N^{-\frac{1}{p+4}}$, the kernel density estimator $\hat{f}$ in* (31) *satisfies*

$$\sup_{f \in \mathcal{F}} E\left[d_2\left(\hat{f}, f\right)^2 \bigg| f\right] = O\left(N^{-4/p+4}\right). \tag{32}$$

*If we assume further (K2), for any $\delta > 0$,*

$$\sup_{f \in \mathcal{F}} P\left(d_\infty\left(\hat{f}, f\right) > N^{-\frac{2}{p+4}+\delta} \bigg| f\right) = o(1) \quad \text{as } N \to \infty. \tag{33}$$

Proposition 3 indicates the convergence rate of the proposed density estimator based on the observation sample size $N$ from the random distribution. The following proposition provides an enhanced form of uniform convergence for the multivariate kernel density estimator under assumption (P1), which is essential for establishing uniform convergence of the representation in the $L^2$ space.

**Proposition 4.** *Assume (D1), (F1'), (K1), (K2) and (P1). For any $R > 0, \delta > 0$,*

$$n \sup_{f \in \mathcal{F}} P\left(d_\infty\left(\hat{f}, f\right) > RN^{-\frac{2}{p+4}+\delta}\right) = o(1) \quad as \ n \to \infty. \tag{34}$$

## G  Additional Asymptotic Convergence Analysis

Convergence analysis of the representations of the process is presented as follows.

**Theorem 3.** *Under assumptions (F1), (T1)-(T2), (E1) and (K1)-(K2), and with $\hat{\nu}$, $\hat{H}$, $\hat{\lambda}_k$, $\hat{\gamma}_k$ as per (14) and (15), for the $k$-th component of the eigen-expansion,*

$$d_2\left(\nu, \hat{\nu}\right) = O_p\left(n^{-1/2}\right), \quad d_\infty\left(\nu, \hat{\nu}\right) = O_p\left(\left(\frac{\log(n)}{n}\right)^{1/2}\right),$$

$$d_2\left(H, \hat{H}\right) = O_p\left(n^{-1/2}\right), \quad d_\infty\left(H, \hat{H}\right) = O_p\left(\left(\frac{\log(n)}{n}\right)^{1/2}\right),$$

$$\left|\lambda_k - \hat{\lambda}_k\right| = O_p\left(n^{-1/2}\right),$$

$$d_2\left(\gamma_k, \hat{\gamma}_k\right) = \sigma_k^{-1} O_p\left(n^{-1/2}\right),$$

$$d_\infty\left(\gamma_k, \hat{\gamma}_k\right) = O_p\left(\lambda_k^{-1}\sigma_k^{-1}n^{-1/2}\right).$$

*Furthermore, if the eigenfunctions correspond to the trigonometric basis, then $\|\gamma_k\|_\infty = O(1)$ and $\|\partial\gamma_k(\theta, s)/\partial s\|_\infty = O(k)$, so that,*

$$\max_{1 \leq k \leq K(n)} \sup_{|\alpha| \leq \alpha_0} d_{SW}\left(m_k(x, \alpha, \tau), \hat{m}_k(x, \alpha, \tau)\right) = O_p(\tau^p \log(n)n^{-1/4}).$$

The following additional assumptions impose additional restrictions on the eigencomponents. They are satisfied for eigenvalue sequences with exponential decay and are needed to show convergence of the $K$-representations to the target densities in Theorem 4. Both conditions involve a sequence of included components $K = K(n)$ with $K \to \infty$ as $n \to \infty$.

(S2) $\left(\sum_{k=1}^{K} \lambda_k^{-1}\sigma_k^{-1}\right)n^{-1/2} = O(1)$.

(S3) $\left(\sum_{k=1}^{K} \lambda_k^{-1}\right)\left(\sum_{k>K} \lambda_k\right) = O(1)$.

The following bound is essential,

$$R(\tau) = \max_{1 \leq k \leq K} \max_{1 \leq i \leq n} C_4\left(\tau, \|X_i\|_\infty, \|\partial X_i(\theta, s)/\partial s\|_\infty, d_\infty\left(X_i, \hat{X}_{ik}\right)\right) \tag{35}$$

$$= O\left(\tau^p \max_{1 \leq k \leq K, 1 \leq i \leq n} e^{2d_\infty\left(X_i, \hat{X}_{ik}\right)}\right). \tag{36}$$

Here $\|X_i\|_\infty$ and $\|\partial X_i(\theta, s)/\partial s\|_\infty$ are bounded, as per (T2). The convergence of $d_\infty\left(X_i, \hat{X}_{ik}\right)$ can be decomposed into two parts, the convergence of $d_\infty\left(X_{ik}, \hat{X}_{ik}\right)$ and that of $d_\infty\left(X_i, X_{ik}\right)$. Both are $O_p(1)$, using assumptions (S2), respectively, (S3).

The convergence rate of the truncated density estimator in (25) is as follows.

**Theorem 4.** *Under assumptions (D1), (F1), (T1)-(T5), (E1), (K1)-(K2), if $K(n) \to \infty$ satisfies conditions (S2)-(S3), then*

$$\max_{1 \leq i \leq n} d_{SW}\left(f_i(\cdot), \hat{f}_i(\cdot, K, \tau)\right) = O_p\left(\tau^{-(\kappa-p)} + \tau^p\left(n^{-1/2}\sum_{k=1}^{K} \sigma_k^{-1} + \left(\sum_{k>K}^{\infty} \lambda_k\right)^{1/2}\right)\right).$$

Under exponentially decaying eigenvalues, one can derive convergence rates that depend on the dimensionality of the domain and the growth behavior of the regularization parameter $\tau$. These rates are obtained for specific choices of $K$ and $\tau$, as stated in Theorem 2, which follows as a corollary of Theorem 4.

# H   Temperature Data Case Study

Our second data illustration involves analyzing maximum and minimum temperature data obtained from the National Centers for Environmental Information, with raw data conveniently accessible at https://www.ncdc.noaa.gov/. This extensive dataset comprehensively covers weather observations spanning the entirety of the United States. We focus on climate stations positioned at major international airports within the 44 largest U.S. cities, since the quality of the data from airport weather stations is superior compared to other general weather stations due to the obvious relevance of weather conditions for the operation of aircraft. The temporal scope of our investigation spans from January 1, 2020, to December 31, 2020, capturing a comprehensive spectrum of climatic conditions throughout the year 2020. In preliminary data processing, we converted the min/max daily temperature into bivariate vectors (min,range), where range=max-min temperature for each day. This simple linear transformation yields bivariate data that are equivalent to the original data but avoids the somewhat awkward triangular support set of the original (min/max) data.

Utilizing kernel density estimation, as illustrated in Figure 10, with a Gaussian density kernel and bandwidth set at 20 for the maximum temperature and 10 for the temperature range, we obtain estimated densities as depicted in Figure 11. Here we utilize five-dimensional representations. Dimension reduction is achieved by employing only five components to represent the infinite-dimensional space while explaining a fraction of variance with respect to the sliced Wasserstein distance that exceeds 0.85.

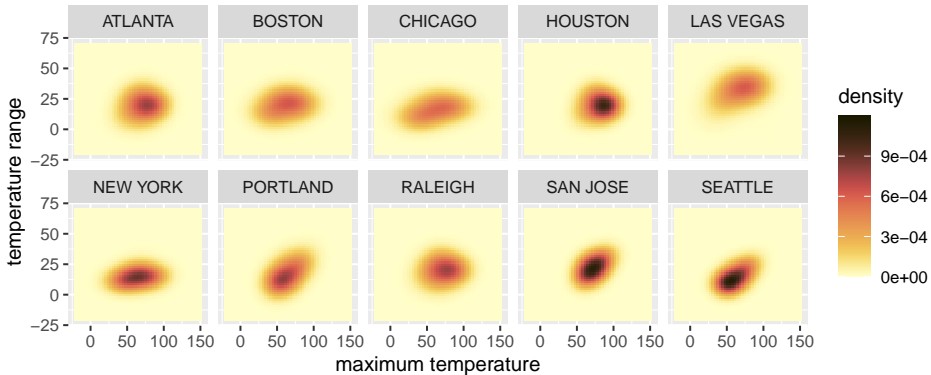

Figure 10: Observed smoothed densities for a few randomly selected airport weather stations.

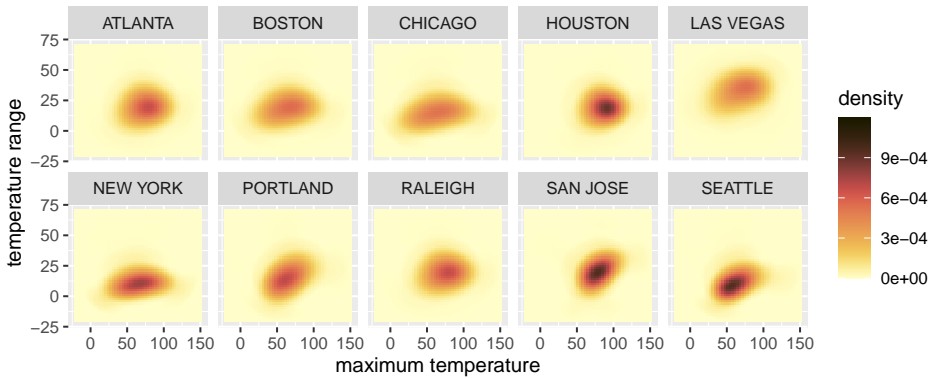

Figure 11: Fitted density surfaces for a few airports with sliced Wasserstein fraction of variance explained at level 0.85 with five components, where the first component contributes 0.41 and the second 0.18.

The first two transformation modes of variation are shown in Figure 12 while scores related to eigenfunctions are presented in Figure 13 and 14. The first model of variation reflects both the intensity of variation in the underlying distribution and the average level of temperature range. A lower score of the first eigenfunction indicates reduced variation and a lower temperature range, while a higher score corresponds to amplified variation, particularly in maximum temperature, and an overall higher temperature range. The second mode of variation reflects the shape of the joint distribution. For smaller scores it reflects a flatter joint distribution, where the temperature range exhibits a lower overall variation and a weaker correlation with the maximum temperature, while higher scores are associated with a more positive correlation between the temperature range and maximum temperature and is accompanied by increased overall variation in the temperature range.

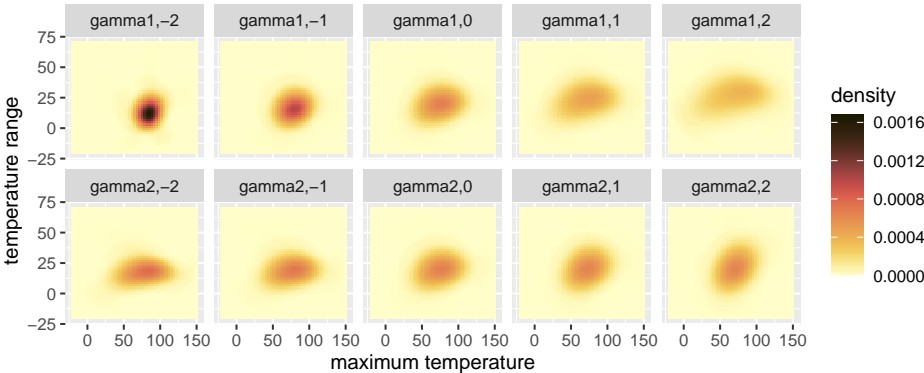

Figure 12: Transformation modes of variation as per (13) for bivariate temperature distributions. The first row corresponds to the first mode of variation and the second row to the second mode of variation, for $\alpha$ varying from $-2$ to 2 as per (17).

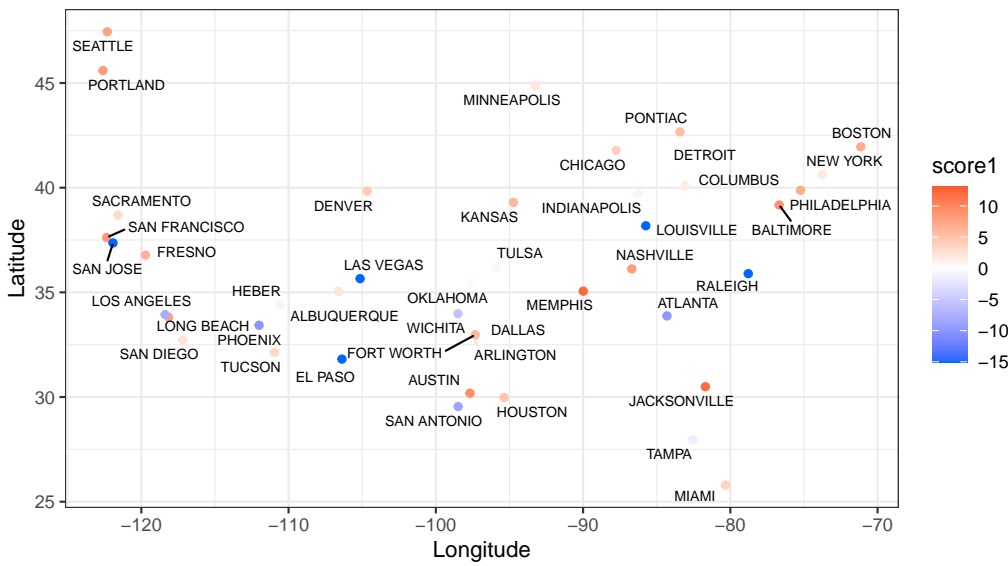

Figure 13: Visualization of the scores for the first eigenfunction.

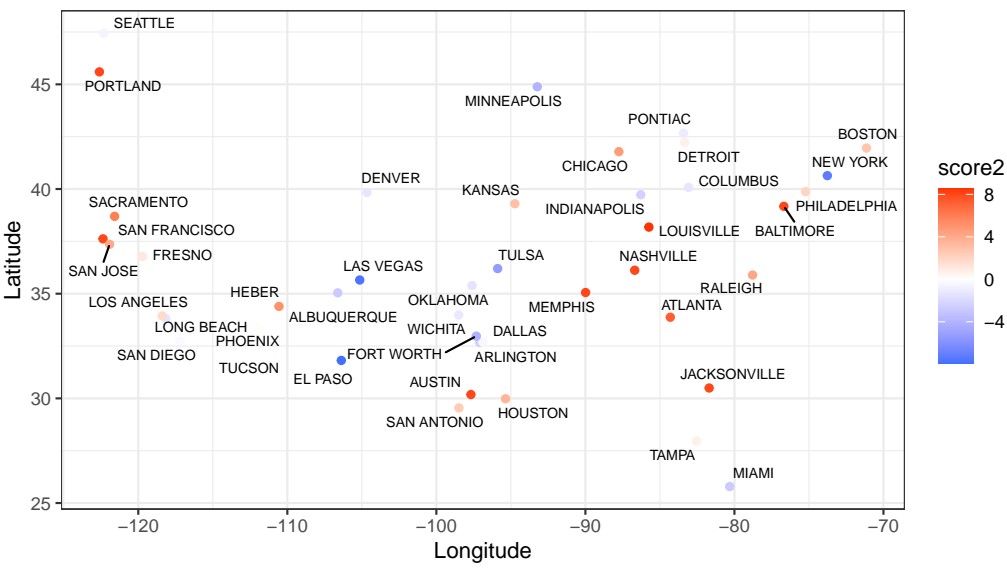

Figure 14: Visualization of the scores corresponding to the second eigenfunction.

Take San Jose and Portland. Both have positive scores for the second eigenfunction, indicating a higher correlation between temperature range and maximum temperature. Nevertheless, San Jose's variance is smaller, attributable to its negative score for the first eigenfunction, in contrast to Portland's positive score On the other side of the spectrum, New York exhibits a negative score for the second eigenfunction, resulting in a flattened shape of the joint distribution. This pattern reflects a weak association between the temperature range and the maximum temperature, accompanied by a lower overall variation in the temperature range. San Jose and Portland's increased correlation between temperature range and maximum temperature likely is due to regional factors, such as maritime influence due to their proximity to the Pacific Ocean, resulting in temperature amplitudes that are relatively smaller in winter compared to summer. New York's weakened association and flatter distribution is also likely linked to its specific geography with relatively little variation in the temperature range.

# I   Computation Resources

All data analyses were performed on a MacBook Pro with an Apple M1 chip and 16GB RAM, running R version 4.3.1 (2023-06-16) under macOS Big Sur. The full analysis for the BLSA dataset and the temperature dataset (presented in the Appendix) each took approximately 5 minutes to complete.

# J   Proofs

## J.1   Proof of Proposition 3

Proof of Proposition 3 follows directly from the proofs of Propositions S2 and S3 in [9].

## J.2   Proof of Proposition 4

Denote the density $f_{D_f}$ on the domain $D$ as $\tilde{f} = f_{D_f} \mathbf{1}_D$. We first show that for any $R > 0, \delta > 0$,

$$n \sup_{f \in \mathcal{F}} P\left(d_\infty\left(\check{f}, \tilde{f}\right) > RN^{-\frac{2}{p+4}+\delta}\Big| f\right) = o(1) \quad \text{as } N \to \infty.$$

By the triangle inequality,

$$d_\infty\left(\check{f}, \tilde{f}\right) \leq d_\infty\left(\check{f}, E(\check{f})\right) + d_\infty\left(E\left(\check{f}\right), \tilde{f}\right).$$

From equation (S.19) in [9] we find

$$\sup_{x \in D} \sup_{f \in \mathcal{F}} \left|\tilde{f}(x) - E\left(\check{f}(x)\right)\right| = O(h^2).$$

By Proposition 9 in [42], there exist constants $L_1$ and $L_2$, which depend only on the VC characteristics of $\mathcal{K}$ in (K2), such that for any $R > 0$ and sequences $a_N$, $L_h = O(h^2)$,

$$P\left(d_\infty(\check{f}, \tilde{f}) > Ra_N\Big| f\right) \leq P\left(d_\infty\left(\check{f}, E\left(\check{f}\right)\right) > Ra_N|f\right) + P\left(d_\infty\left(E\left(\check{f}\right), \tilde{f}\right) > Ra_N\Big| f\right)$$

$$\leq L_1 \exp\{-L_2 Nh^p R^2 a_N^2\} + I\{L_h > Ra_N\},$$

where $I$ is the indicator function. Let $a_N = N^{-2/(p+4)+\delta}$ for any $\delta > 0$, and $h \sim N^{-\frac{1}{p+4}}$. Thus

$$\sup_{f \in \mathcal{F}} P\left(d_\infty\left(\check{f}, \tilde{f}\right) > Ra_N\Big| f\right) \leq L_1 \exp\{-L_2 R^2 N^{2\delta}\}.$$

Under assumption (P1), $N(n) \sim n^r$, for a constant $r > 1 + p/4$, therefore

$$n \sup_{f \in \mathcal{F}} P\left(d_\infty\left(\check{f}, \tilde{f}\right) > Ra_N\Big| f\right) \leq nL_1 \exp\{-L_2 R^2 n^{2\delta r}\} = o(1) \quad \text{as } n \to \infty.$$

Next, we show the convergence of the truncated density $\hat{f}$, where the target is $f(x) = \tilde{f}(x)/\int_D \tilde{f}(u)du, x \in D$. Note that when $N$ is large enough we have

$$\int_D \tilde{f}(u)du \geq \frac{|D|}{M_0}, \quad \int_D \check{f}(u)du \geq \frac{|D|}{2M_0},$$

whence

$$d_\infty\left(\hat{f}, f\right) \leq d_\infty\left(\frac{\check{f}(x)}{\int_D \check{f}(u)du}, \frac{\tilde{f}(x)}{\int_D \check{f}(u)du}\right) + d_\infty\left(\frac{\tilde{f}(x)}{\int_D \check{f}(u)du}, \frac{\tilde{f}(x)}{\int_D \tilde{f}(u)du}\right)$$

$$\leq \frac{2M_0}{|D|}d_\infty(\check{f}, \tilde{f}) + \frac{2M_0^3}{|D|}d_\infty(\check{f}, \tilde{f}).$$

It follows that for any $R > 0 \, and \, \delta > 0$,

$$n \sup_{f \in \mathcal{F}} P\left(d_\infty\left(\hat{f}, f\right) > RN^{-\frac{2}{p+4}+\delta}\Big| f\right) = o(1) \quad \text{as } n \to \infty.$$

### J.3 Proof of Proposition 2

**Lemma 1.** *Under (D1) and (F1), there exists a constant $A$ such that,*

$$\sup_{f \in \mathcal{F}} |\mathcal{R}(f)(\theta_1, s) - \mathcal{R}(f)(\theta_2, s)| \le A\|\theta_1 - \theta_2\|_2, \quad \text{for all } \theta_1, \theta_2 \in \Theta.$$

*Proof of Lemma 1.* We first show the result for $p = 2$. Write $\theta(\alpha) = (\cos\alpha, \sin\alpha)^T \in \mathbb{S}^1$ where $\alpha \in [0, 2\pi]$ and $x = (x_1, x_2)^T \in \mathbb{R}^2$. Let $\Pi_\Delta$ be the two-dimensional rotation operator that employs the angle $\Delta$ and $f_{\Pi_\Delta} = f(\Pi_\Delta \circ x)$. Note that for all $z$,

$$\mathcal{R}(f_{\Pi_\Delta})(\theta(\alpha), z) = \mathcal{R}(f)(\theta(\alpha + \Delta), z).$$

Calculating the derivative with respect to $\alpha$,

$$\mathcal{R}(f)(\theta(\alpha + \Delta), z) - \mathcal{R}(f)(\theta(\alpha), z) = \mathcal{R}(f_{\Pi_\Delta} - f)(\theta(\alpha), z). \tag{37}$$

Note that the two-dimensional rotation operator can be represented as a rotation matrix,

$$
\begin{aligned}
\Pi_\Delta \circ x &= \begin{pmatrix} \cos\Delta & -\sin\Delta \\ \sin\Delta & \cos\Delta \end{pmatrix} \circ \begin{pmatrix} x_1 \\ x_2 \end{pmatrix} \\
&= \begin{pmatrix} x_1 \cos\Delta - x_2 \sin\Delta \\ x_1 \sin\Delta + x_2 \cos\Delta \end{pmatrix}
\end{aligned}
\tag{38}
$$

Then,

$$
\begin{aligned}
f_{\Pi_\Delta}(x_1, x_2) - f(x_1, x_2) &= f(x_1 \cos\Delta - x_2 \sin\Delta, x_1 \sin\Delta + x_2 \cos\Delta) - f(x_1, x_2) \\
&= \frac{\partial f}{\partial x_1}(x_1^*, x_2^*)(x_1(\cos\Delta - 1) - x_2 \sin\Delta) \\
&\quad + \frac{\partial f}{\partial x_2}(x_1^*, x_2^*)(x_1 \sin\Delta + x_2(\cos\Delta - 1)),
\end{aligned}
$$

where $x_1^* = (1 - c_0)x_1 + c_0(x_1 \cos\Delta - x_2 \sin\Delta)$ and $x_2^* = (1 - c_0)x_2 + c_0(x_1 \sin\Delta + x_2 \cos\Delta)$ for a $c_0 \in (0, 1)$. Thus,

$$\lim_{\Delta \to 0} \frac{f_{\Pi_\Delta}(x_1, x_2) - f(x_1, x_2)}{\Delta} = -x_2 \frac{\partial f}{\partial x_1}(x_1, x_2) + x_1 \frac{\partial f}{\partial x_2}(x_1, x_2). \tag{39}$$

Combining (37) and (39), we have

$$
\begin{aligned}
\frac{\partial \mathcal{R}(f(x_1, x_2))}{\partial \alpha}(\theta(\alpha), z) &= \lim_{\Delta \to 0} \frac{\mathcal{R}(f(x_1, x_2))(\theta(\alpha + \Delta), z) - \mathcal{R}(f(x_1, x_2))(\theta(\alpha), z)}{\Delta} \\
&= \lim_{\Delta \to 0} \frac{\mathcal{R}(f_{\Pi_\Delta}(x_1, x_2) - f(x_1, x_2))(\theta(\alpha), z)}{\Delta} \\
&= \mathcal{R}\left[\left(x_1 \frac{\partial f}{\partial x_2} - x_2 \frac{\partial f}{\partial x_1}\right)(x_1, x_2)\right](\theta(\alpha), z).
\end{aligned}
$$

Under (D1) and (F1), there exists a constant $A_0$ such that

$$\left|\frac{\partial \mathcal{R}(f)}{\partial \alpha}(\theta(\alpha), z)\right| \le A_0.$$

Let $\theta_1 = \theta(\alpha_1)$ and $\theta_2 = \theta(\alpha_2) \in \mathbb{S}^2$, $\alpha_1, \alpha_2 \in [0, 2\pi]$, where we note that $\|\theta_1 - \theta_2\|_2 = |2\sin((\alpha_1 - \alpha_2)/2)| \gtrsim |\alpha_1 - \alpha_2|$ when $|\alpha_1 - \alpha_2| \to 0$. Thus, for any $\theta_1, \theta_2 \in \mathbb{S}^1, z \in \mathbb{R}, f \in \mathcal{F}$,

$$|\mathcal{R}(f)(\theta_1, z) - \mathcal{R}(f)(\theta_2, z)| \le A_0 |\alpha_1 - \alpha_2| \lesssim 2A_0 \|\theta_1 - \theta_2\|_2, \quad \text{as } \|\theta_1 - \theta_2\|_2 \to 0.$$

Next, we provide the proof for $p = 3$.
Let $\alpha \in [0, 2\pi]$, $\beta \in [0, \pi]$, $\theta(\alpha, \beta) = (\sin\beta \cos\alpha, \sin\beta \sin\alpha, \cos\beta)^T \in \mathbb{S}^2$
and $x = (x_1, x_2, x_3)^T \in \mathbb{R}^3$. Noting that

$$\frac{\partial R(f(x_1, x_2, x_3))}{\partial \alpha}(\theta(\alpha, \beta), z)$$

$$= -(\sin\alpha \sin\beta)\mathcal{R}\left[\frac{\partial f}{\partial x_1}(x_1, x_2, x_3)\right](\theta(\alpha, \beta), z) + (\cos\alpha \sin\beta)\mathcal{R}\left[\frac{\partial f}{\partial x_2}(x_1, x_2, x_3)\right](\theta(\alpha, \beta), z)$$

$$\frac{\partial R\left(f(x_1, x_2, x_3)\right)}{\partial \beta}(\theta(\alpha, \beta), z)$$

$$= -(\sin \alpha \cos \beta)\mathcal{R}\left[\frac{\partial f}{\partial x_1}(x_1, x_2, x_3)\right](\theta(\alpha, \beta), z) + (\cos \alpha \cos \beta)\mathcal{R}\left[\frac{\partial f}{\partial x_2}(x_1, x_2, x_3)\right](\theta(\alpha, \beta), z)$$

$$- (\sin \beta)\mathcal{R}\left[\frac{\partial f}{\partial x_3}(x_1, x_2, x_3)\right](\theta(\alpha, \beta), z),$$

under (D1) and (F1), there exists a constant $A_0$ such that

$$\left|\frac{\partial \mathcal{R}(f)}{\partial \alpha}(\theta(\alpha, \beta), z)\right| \le 2A_0|\sin \beta|, \quad \left|\frac{\partial \mathcal{R}(f)}{\partial \beta}(\theta(\alpha, \beta), z)\right| \le 3A_0.$$

Let $\theta_1 = \theta(\alpha_1, \beta_1)$ and $\theta_2 = \theta(\alpha_2, \beta_2) \in \mathbb{S}^3$, $\alpha_1, \alpha_2 \in [0, 2\pi]$ and $\beta_1, \beta_2 \in [0, \pi]$. Since $\|\theta_1 - \theta_2\|_2 = \sqrt{2 - 2\left(\sin \beta_1 \sin \beta_2 \cos(\alpha_2 - \alpha_1) + \cos \beta_1 \cos \beta_2\right)} \gtrsim |\sin \beta_1||\alpha_2 - \alpha_1| + |\beta_2 - \beta_1|$ when $\|\theta_2 - \theta_1\|_2 \to 0$, for any $\theta_1, \theta_2 \in \mathbb{S}^2$, $z \in \mathbb{R}$, $f \in \mathcal{F}$, as $\|\theta_1 - \theta_2\|_2 \to 0$,

$$|\mathcal{R}(f)(\theta_1, z) - \mathcal{R}(f)(\theta_2, z)| \le 2A_0|\sin \beta_1||\alpha_2 - \alpha_1| + 3A_0|\beta_2 - \beta_1|$$
$$\lesssim \|\theta_2 - \theta_1\|_2.$$

Following these arguments, it is clear how the proof can be extended to the multivariate case when $p > 3$, and we omit the details. $\qquad\square$

The analytical tools in Section 2.2 provide the following property for the sliced density function, which implies that the sliced univariate density function adheres to the domain assumption (A1) in [35]. This observation is basic for our subsequent exploration of multivariate density transformation.

**Lemma 2.** *Under assumptions (D1) and (F1), $\mathcal{R}(f)(\theta, \cdot)$, $f \in \mathcal{F}$, is $\kappa$-differentiable, $\kappa \ge p + 1$, and there exists a constant $C_0$ such that*

$$\sup_{f \in \mathcal{F}} \|\mathcal{R}(f)\|_\infty < C_0, \quad \sup_{f \in \mathcal{F}} \|1/\mathcal{R}(f)\|_\infty < C_0, \quad \sup_{f \in \mathcal{F}} \left\|\frac{\partial^\kappa \mathcal{R}(f)}{\partial z}(\theta, z)\right\|_\infty < C_0.$$

*Proof of Lemma 2.* From assumption (F1) which states that the density $f$ is uniformly bounded above and below, it follows that

$$\|\mathcal{R}(f)\|_\infty \le M_0|D|, \quad \|1/\mathcal{R}(f)\|_\infty \le M_0|D|,$$

where $|D|$ represents the Lebesgue measure of the domain set $D$. From (S1) in [9] and assumption (F1), there exists an constant $C_0 \ge M_0|D|$ such that

$$\left\|\frac{\partial^k \mathcal{R}(f)}{\partial z}(\theta, z)\right\|_\infty \le C_0.$$

Note that the constant $C_0$ is chosen independently of the function $f$, which completes the proof of the proposition.

$\qquad\square$

*Proof of Proposition 2.* Denote the slice-wise density transformation $\widetilde{\psi}$ as

$$\widetilde{\psi}(h)(\theta, s) = \psi(h(\theta, \cdot))(s), \quad h \in \mathcal{R}(\mathcal{F})$$

and the slice-wise inverse transformation $\widetilde{\psi}^{-1}$ as

$$\widetilde{\psi}^{-1}(\zeta)(\theta, z) = \psi^{-1}(\zeta(\theta, \cdot))(z), \quad \zeta \in \Psi(\mathcal{F}). \tag{40}$$

The multivariate density transformation in (6) is $\Psi(f)(\theta, s) = \widetilde{\psi} \circ \mathcal{R}(f)(\theta, s)$. From assumption (D1) and Proposition 2 in [9], for any $f_1, f_2 \in \mathcal{F}$, there exists a constant $A_1$ such that

$$d_2\left(\mathcal{R}(f_1), \mathcal{R}(f_2)\right) \le A_1 d_2(f_1, f_2),$$
$$d_\infty\left(\mathcal{R}(f_1), \mathcal{R}(f_2)\right) \le A_1 d_\infty(f_1, f_2).$$

Furthermore, from Proposition 2, there exists a constant $A_2$ such that

$$\|\mathcal{R}(f)\|_\infty \leq A_2, \quad \|1/\mathcal{R}(f)\|_\infty \leq A_2, \quad \left\|\frac{\partial \mathcal{R}(f)(\theta, z)}{\partial z}\right\|_\infty \leq A_2, \quad \text{for all } f \in \mathcal{F}.$$

From (L1), for any $f_1, f_2 \in \mathcal{F}$, there exists a constant $A_3$ such that

$$
\begin{aligned}
d_2\left(\widetilde{\psi} \circ \mathcal{R}(f_1), \widetilde{\psi} \circ \mathcal{R}(f_2)\right) &= \left(\int_{\theta \in \Theta} \int_{u \in \mathcal{S}} \left(\widetilde{\psi} \circ \mathcal{R}(f_1)(\theta, u) - \widetilde{\psi} \circ \mathcal{R}(f_2)(\theta, u)\right)^2\right)^{1/2} \\
&\leq \left(\int_{\theta \in \Theta} A_3^2 d_2^2\left(\mathcal{R}(f_1)(\theta, \cdot), \mathcal{R}(f_2)(\theta, \cdot)\right)\right)^{1/2} \\
&= A_3 d_2(\mathcal{R}(f_1), \mathcal{R}(f_2)).
\end{aligned}
$$

Similarly,

$$d_\infty\left(\widetilde{\psi} \circ \mathcal{R}(f_1), \widetilde{\psi} \circ \mathcal{R}(f_2)\right) \leq A_3 d_\infty(\mathcal{R}(f_1), \mathcal{R}(f_2)),$$

which proves the existence of $C_1 = A_1 A_3$ in (T1). Furthermore from (L2), there exists a constant $A_4$ such that

$$\sup_{f \in \mathcal{F}} \left\|\left(\widetilde{\psi} \circ \mathcal{R}\right)(f)\right\|_\infty \leq A_4, \quad \sup_{f \in \mathcal{F}} \left\|\frac{\partial \left(\widetilde{\psi} \circ \mathcal{R}\right)(f)}{\partial z}(\theta, z)\right\|_\infty \leq A_4.$$

From Lemma 1 and (L1), it follows that there also exists a constant $A_5$ such that for any $f \in \mathcal{F}$,

$$
\begin{aligned}
\sup_s \left|\left(\widetilde{\psi} \circ \mathcal{R}\right)(f)(\theta_1, s) - \left(\widetilde{\psi} \circ \mathcal{R}\right)(f)(\theta_2, s)\right| &\leq \tilde{C}_0 \sup_z |\mathcal{R}(f)(\theta_1, z) - \mathcal{R}(f)(\theta_2, z)| \\
&\leq \tilde{C}_0 A_5 \|\theta_1 - \theta_2\|_2.
\end{aligned}
$$

This shows the existence of $C_2$ in (T2). Assumption (T3) is supported by the inverse transformation formula (40) and the injectivity of the Radon transform on rapidly decreasing functions, as established in Theorem 2.4 of [21]. The regularized inverse (7) is $\Psi_\tau^{-1} = \mathcal{R}_\tau^{-1} \circ \widetilde{\psi}$. From Corollary 1 of [9], there exists a constant depending on $C_3(\tau) = O(\tau^{-(\kappa-p)})$ as $\tau \to \infty$ such that

$$
\begin{aligned}
d_\infty(\Psi_\tau^{-1} \circ \Psi(f), f) &= d_\infty\left(\mathcal{R}_\tau^{-1} \circ \widetilde{\psi}^{-1} \circ \widetilde{\psi} \circ \mathcal{R}(f), f\right) \\
&= d_\infty\left(\mathcal{R}_\tau^{-1} \circ \mathcal{R}(f), f\right) \\
&\leq C_3(\tau),
\end{aligned}
$$

which shows the existence of $C_3$ in (T4). Furthermore, there exists a constant $A_6(\tau) = O(\tau^p)$ as $\tau \to \infty$, such that

$$
\begin{aligned}
d_\infty\left(\Psi_\tau^{-1}(\zeta_1), \Psi_\tau^{-1}(\zeta_2)\right) &= d_\infty\left(\mathcal{R}_\tau^{-1} \circ \widetilde{\psi}^{-1}(\zeta_1), \mathcal{R}_\tau^{-1} \circ \widetilde{\psi}^{-1}(\zeta_2)\right) \\
&\leq A_6(\tau) d_2\left(\widetilde{\psi}^{-1}(\zeta_1), \widetilde{\psi}^{-1}(\zeta_2)\right).
\end{aligned}
\tag{41}
$$

From assumption (L3), there exist constants $A_7(\|\zeta_1\|_\infty, \|\partial \zeta_1(\theta, s)/\partial s\|_\infty)$ and $A_8(d_\infty(\zeta_1, \zeta_2))$,

$$
\begin{aligned}
d_2(\widetilde{\psi}^{-1}(\zeta_1), \widetilde{\psi}^{-1}(\zeta_2)) &= \left(\int_{\theta \in \mathcal{S}^{p-1}} \int_z \left(\psi^{-1}(\zeta_1(\theta, \cdot))(z) - \psi^{-1}(\zeta_2(\theta, \cdot))(z)\right)^2\right)^{1/2} \\
&\leq \left(\int_{\theta \in \mathcal{S}^{p-1}} A_7^2 A_8^2 d_2\left(\zeta_1(\theta, \cdot), \zeta_2(\theta, \cdot)\right)^2\right)^{1/2} \\
&= A_7 A_8 d_2(\zeta_1, \zeta_2).
\end{aligned}
\tag{42}
$$

Combining (41) and (42), we have

$$d_\infty(\Psi_\tau^{-1}(\zeta_1), \Psi_\tau^{-1}(\zeta_2)) \leq A_6(\tau) A_7(\|\zeta_1\|_\infty, \|\partial \zeta_1(\theta, s)/\partial s\|_\infty) A_8(d_\infty(\zeta_1, \zeta_2)) d_2(\zeta_1, \zeta_2),$$

which proves the existence of $C_4 = A_6 A_7 A_8$ in (T5). From Proposition 2 of [35], when using the LQD transformation $\psi_Q$,

$$A_7 = O\left(\|\partial\zeta_1(\theta,s)/\partial s\|_\infty e^{6\|\zeta_1\|_\infty}\right), \quad A_8 = O\left(e^{2d_\infty(\zeta_1,\zeta_2)}\right).$$

Thus, when using the RLQD transformation $\Psi_Q$, we have

$$C_4(\tau,\|\zeta_1\|_\infty,\|\partial\zeta_1(\theta,s)/\partial s\|_\infty, d_\infty(\zeta_1,\zeta_2)) \sim \tau^p\|\partial\zeta_1(\theta,s)/\partial s\|_\infty e^{6\|\zeta_1\|_\infty}e^{2d_\infty(\zeta_1,\zeta_2)}.$$

For the LH transformation $\psi_H$,

$$A_7 = O\left(e^{2\|\zeta_1\|_\infty}\right), \quad A_8 = O\left(e^{d_\infty(\zeta_1,\zeta_2)}\right).$$

Thus, when using the RLH transformation $\Psi_H$, we have

$$C_4(\tau,\|\zeta_1\|_\infty,\|\partial\zeta_1(\theta,s)/\partial s\|_\infty, d_\infty(\zeta_1,\zeta_2)) \sim \tau^p e^{2\|\zeta_1\|_\infty}e^{d_\infty(\zeta_1,\zeta_2)}.$$

$\square$

## J.4 Proof of Proposition 1

*Proof.* From assumption (T2),

$$\|\Psi(f)\|_\infty \le C_2, \|\partial s\Psi(f)(\theta,s)/\partial s\|_\infty \le C_2.$$

From the fact that $f \in \mathcal{F}$ and $\zeta$ is a continuous function on $L^2(\Theta \times \mathcal{S})$,

$$d_\infty(\Psi(f),\zeta) \le \|\Psi(f)\|_\infty + \|\zeta\|_\infty < \infty.$$

Thus

$$C_4\left(\tau,\|\Psi(f)\|_\infty,\|\partial\Psi(f)(\theta,s)/\partial s\|_\infty, d_\infty(\Psi(f),\zeta)\right) < \infty.$$

From assumption (T5),

$$d_\infty\left(\Psi_\tau^{-1}(\Psi(f)),\Psi_\tau^{-1}(\zeta)\right) \le C_4 d_2\left(\Psi(f),\zeta\right),$$

and from assumption (T4),

$$d_\infty\left(f,\Psi_\tau^{-1}(\Psi(f))\right) \le C_3.$$

Then

$$d_\infty\left(f,\Psi_\tau^{-1}(\zeta)\right) \le C_3 + C_4 d_2\left(\Psi(f),\zeta\right).$$

From Assumption (D1), we have

$$d_2\left(f,\Psi_\tau^{-1}(\zeta)\right) \le C_3 + C_4 d_2\left(\Psi(f),\zeta\right).$$

From Lemma S3 in [9] and assumption (D1), it follows that

$$d_{SW}\left(f,\Psi_\tau^{-1}(\zeta)\right) = O\left(C_3 + C_4 d_2\left(\Psi(f),\zeta\right)\right).$$

$\square$

## J.5 Proof of Theorem 3

**Lemma 3.** *Assume (T2) and let $\tilde{\nu}$ as per (22), then*

$$d_\infty(\tilde{\nu},\nu) = O_p\left((\log n/n)^{1/2}\right). \tag{43}$$

*Proof of Lemma 3.* As $\mathcal{S} \subset \mathbb{R}$ is a compact interval, one can find a grid $\mathcal{T}_\mathcal{S} = \{s_1, ..., s_a\}$ with $|\mathcal{T}_\mathcal{S}| = O(n)$ such that for any $s \in \mathcal{S}$, there exists a point $s_m \in \mathcal{T}_\mathcal{S}$ satisfying $|s - s_m| < 1/n$. Similarly, one can find a set $\mathcal{T}_\Theta = \{\theta_1, ..., \theta_b\}$ such that for any $\theta \in \Theta$, there exists a point $\theta_m \in \mathcal{T}_\Theta$ satisfying $\|\theta - \theta_m\|_2 < 1/n$. The size of the set $|\mathcal{T}_\Theta|$ is determined by the covering number of $\Theta$. From Lemma 5.7 and Example 5.8 in [46], one can choose $\mathcal{T}_\Theta$ such that its size satisfies

$|\mathcal{T}_\Theta| = O(n^p)$. Forming the product of $\mathcal{T}_\mathcal{S}$ and $\mathcal{T}_\Theta$ leads to the grid $\mathcal{T} = \mathcal{T}_\Theta \times \mathcal{T}_\mathcal{S}$ on $\Theta \times \mathcal{S}$ with $|\mathcal{T}| = O(n^{p+1})$.

From the smoothness assumption in (T2),

$$\sup_{\theta \in \Theta, s \in \mathcal{S}} |\tilde{\nu}(\theta, s) - \nu(\theta, s)| = \sup_{\theta, s \in \mathcal{T}} |\tilde{\nu}(\theta, s) - \nu(\theta, s)| + O(1/n).$$

Under (T2), $\|X_i(\theta, s)\|_\infty \leq C_2$. Applying Bernstein inequality for a fixed $(\theta, s) \in \mathcal{T}$, for any $R > 0$,

$$P\left(|\tilde{\nu}(\theta, s) - \nu(\theta, s)| \geq R\left(\frac{\log(n)}{n}\right)^{1/2}\right) = P\left(\left|\frac{1}{n}\sum_{i=1}^n X_i(\theta, s) - \nu(\theta, s)\right| \geq R\left(\frac{\log(n)}{n}\right)^{1/2}\right)$$

$$\leq 2\exp\left(-\frac{3(n\log n)R^2}{6nC_2^2 + 2C_2 R(n\log n)^{1/2}}\right).$$

Then

$$P\left(\sup_{\theta, s \in \mathcal{T}} |\tilde{\nu}(\theta, s) - \nu(\theta, s)| \geq R\left(\frac{\log(n)}{n}\right)^{1/2}\right) \leq 2n^{p+1}\exp\left(-\frac{3(n\log n)R^2}{6nC_2^2 + 2C_2 R(n\log n)^{1/2}}\right)$$

$$= 2n^{p+1-R^*}, \tag{44}$$

where $R^* = \frac{3R^2}{6C_2^2 + 2C_2 R(\log n/n)^{1/2}}$. For a fixed dimension $p$, choose a large enough $R$ such that $R^* > p + 1$. Then

$$\sup_{\theta \in \Theta, s \in \mathcal{S}} |\tilde{\nu}(\theta, s) - \nu(\theta, s)| = O_p\left(\left(\frac{\log n}{n}\right)^{1/2}\right). \tag{45}$$

$\square$

*Proof of Theorem 3.* Assumption (T2) implies that $E\|X\|_2^2 < \infty$. Denote by $\tilde{\nu}, \tilde{G}$ the population mean function and covariance function as per (22). Using Theorems 3.9 and 4.2 in [8], it follows that

$$d_2(\nu, \tilde{\nu}) = O_p(n^{-1/2}), \quad d_2(G, \tilde{G}) = O_p(n^{-1/2}).$$

From Proposition 3,

$$\sup_{f \in \mathcal{F}} E\left[d_2\left(f, \check{f}\right)^2 \Big| f\right] = O\left(N^{-\frac{4}{p+4}}\right).$$

The convergence rate of $d_2(\tilde{\nu}, \hat{\nu})$ is

$$P(\{d_2(\tilde{\nu}, \hat{\nu}) > RN^{-\frac{2}{p+4}}\}) \leq P\left(\frac{1}{n}\sum_{i=1}^n d_2(X_i, \check{X}_i) > RN^{-\frac{2}{p+4}}\right)$$

$$\leq \frac{\frac{1}{n}\sum_{i=1}^n E(d_2(X_i, \check{X}_i))}{RN^{-\frac{2}{p+4}}}$$

$$\leq \frac{\frac{1}{n}\sum_{i=1}^n \sqrt{E\left(d_2(X_i, \check{X}_i)^2\right)}}{RN^{-\frac{2}{p+4}}}$$

$$\leq \frac{\frac{1}{n}\sum_{i=1}^n C_1\sqrt{E\left(d_2(f_i, \check{f}_i)^2\right)}}{RN^{-\frac{2}{p+4}}}$$

$$= O\left(\frac{1}{R}\right),$$

which indicates that $d_2(\tilde{\nu}, \hat{\nu}) = O_p(N^{-\frac{2}{p+4}})$, then $d_2(\nu, \hat{\nu}) = O_p(n^{-1/2} + N^{-\frac{2}{p+4}})$. Similarly, we have $d_2(H, \hat{H}) = O_p(n^{-1/2} + N^{-\frac{2}{p+4}})$. In terms of uniform convergence, Lemma 3 provides the uniform convergence for $\tilde{\nu}$ and similarly for $\tilde{H}$,

$$d_\infty(\tilde{\nu}, \nu) = O_p\left(\left(\frac{\log n}{n}\right)^{1/2}\right), \quad d_\infty(\tilde{H}, H) = O_p\left(\left(\frac{\log n}{n}\right)^{1/2}\right).$$

From Proposition 3, for any $R > 0, \delta > 0$,

$$P\left(d_\infty(\tilde{\nu}, \hat{\nu}) > RN^{-\frac{2}{p+4}+\delta}\right) \leq n \max_{1 \leq i \leq n} P(d_\infty(f_i, \check{f}) > C_1^{-1}RN^{-\frac{2}{p+4}+\delta})$$

$$= o(1).$$

Thus $d_\infty(\hat{\nu}, \nu) = O_p\left(\left(\frac{\log n}{n}\right)^{1/2} + N^{-\frac{2}{p+4}+\delta}\right)$ and similarly,

$d_\infty(\hat{H}, H) = O_p\left(\left(\frac{\log n}{n}\right)^{1/2} + N^{-\frac{2}{p+4}+\delta}\right)$. Based on Lemma (4.2) and (4.3) of [8], it follows that

$$|\lambda_k - \hat{\lambda}_k| = O_p(n^{-1/2} + N^{-\frac{2}{p+4}}),$$

$$d_2(\gamma_k, \hat{\gamma}_k) = \sigma_k^{-1} O_p(n^{-1/2} + N^{-\frac{2}{p+4}}).$$

In terms of the uniform convergence rate of $\gamma_k$, employing the techniques in Lemma 1 of [33] and the proof of Corollary 1 of [47],

$$d_\infty\left(\hat{\lambda}_k\gamma_k, \hat{\lambda}_k\hat{\gamma}_k\right) \leq d_\infty\left(\lambda_k\gamma_k, \hat{\lambda}_k\hat{\gamma}_k\right) + d_\infty\left(\lambda_k\gamma_k, \hat{\lambda}_k\gamma_k\right)$$

$$\lesssim d_\infty\left(H, \hat{H}\right) + d_2\left(\gamma_k, \hat{\gamma}_k\right)\|H\|_\infty + \left|\lambda_k - \hat{\lambda}_k\right|\|\gamma_k\|_\infty.$$

Thus,

$$d_\infty\left(\gamma_k, \hat{\gamma}_k\right) = O_p\left(\lambda_k^{-1}\sigma_k^{-1}n^{-1/2}\right).$$

$\square$

## J.6  Proof of Theorem 1

**Lemma 4.** *Under (D1), (F1) and (T2), there exists a constant $C$ such that*

$$\|\nu\|_\infty \leq C, \quad \|\partial\nu(\theta, s)/\partial s\|_\infty \leq C, \tag{46}$$

$$\|\gamma_k\|_\infty \leq C\lambda_k^{-1}, \quad \|\partial\gamma_k(\theta, s)/\partial s\|_\infty \leq C\lambda_k^{-1}. \tag{47}$$

*Proof of Lemma 4.* The derivation of (46) follows directly from (T2). Note that

$$\gamma_k(\theta, s) = \lambda_k^{-1}\int_{\Theta \times \mathcal{S}} H(\theta, s, \alpha, t)\gamma_k(\alpha, t)d\alpha dt.$$

(T2) and the fact that $\|\gamma_k\|_2 = 1$ then imply (47). $\square$

*Proof of Theorem 1.* Assume $K$ is fixed. Then from Lemma 4,

$$\|X_{k,\alpha}\|_\infty \leq \|\nu\|_\infty + \alpha_0\sqrt{\lambda_1}\|\gamma_k\|_\infty \leq C\left(1 + \alpha_0\sqrt{\lambda_1}\lambda_K^{-1}\right). \tag{48}$$

Similarly,

$$\|\partial X_{k,\alpha}(\theta, s)/\partial s\|_\infty \leq C\left(1 + \alpha_0\sqrt{\lambda_1}\lambda_K^{-1}\right). \tag{49}$$

Employing Theorem 3,

$$d_\infty(X_{k,\alpha}, \hat{X}_{k,\alpha}) \leq d_\infty(\nu, \hat{\nu}) + \alpha_0\left(\sqrt{\hat{\lambda}_1}d_\infty(\gamma_k, \hat{\gamma}_k) + \|\gamma_k\|_\infty\left|\sqrt{\lambda_k} - \sqrt{\hat{\lambda}_k}\right|\right)$$

$$= O_p(\sigma_K^{-1}\lambda_K^{-1}n^{-\frac{1}{2}}).$$

Here we use the fact that

$$\left|\sqrt{\lambda_k} - \sqrt{\hat{\lambda}_k}\right| = \frac{|\lambda_k - \hat{\lambda}_k|}{\sqrt{\lambda_k} - \sqrt{\hat{\lambda}_k}} \sim \lambda_k^{-\frac{1}{2}}|\lambda_k - \hat{\lambda}_k|, \quad \text{as } n \to \infty.$$

For the $L^2$ convergence,

$$d_2(X_{k,\alpha}, \hat{X}_{k,\alpha}) \le d_2(\nu, \hat{\nu}) + \alpha_0 \left( \sqrt{\lambda_1} d_2(\gamma_k, \hat{\gamma}_k) + \left| \sqrt{\lambda_k} - \sqrt{\hat{\lambda}_k} \right| \right)$$

$$= O_p(\sigma_K^{-1} n^{-1/2}).$$

From assumption (T4),

$$d_\infty \left( m_k(\alpha, \Psi, \tau), \hat{m}_k(\alpha, \Psi, \tau) \right)$$
$$\le C_4(\tau, \|X_{k,\alpha}\|_\infty, \|\partial X_{k,\alpha}(\theta, s)/\partial s\|_\infty, d_\infty(X_{k,\alpha}, \hat{X}_{k,\alpha})) d_2(X_{k,\alpha}, \hat{X}_{k,\alpha}).$$

Case 1: When $K$ is fixed,

$$\max\{\|X_{k,\alpha}\|_\infty, \|\partial X_{k,\alpha}(\theta, s)/\partial s\|_\infty, d_\infty(X_{k,\alpha}, \hat{X}_{k,\alpha})\} = O_p(1).$$

Writing $S(\tau, K)$ as per (19), by (T5) it follows that $S(\tau, K) = O(\tau^p)$. Thus

$$d_\infty \left( m_k(\alpha, \Psi, \tau), \hat{m}_k(\alpha, \Psi, \tau) \right) = O_p(\tau^p n^{-\frac{1}{2}}).$$

From Lemma S3 in [9] and assumption (D1),

$$d_{SW} \left( m_k(\alpha, \Psi, \tau), \hat{m}_k(\alpha, \Psi, \tau) \right) = O_p(\tau^p n^{-\frac{1}{2}}).$$

Case 2: When $K = K(n) \to \infty$,

$$d_\infty(X_{k,\alpha}, \hat{X}_{k,\alpha}) = O_p(1).$$

From assumption (T5), (48) and (49),

$$d_\infty \left( m_k(\alpha, \Psi, \tau), \hat{m}_k(\alpha, \Psi, \tau) \right)$$
$$\le C_4 \left( \tau, \|X_{k,\alpha}\|_\infty, \|\partial X_{k,\alpha}(\theta, s)/\partial s\|_\infty, d_\infty(X_{k,\alpha}, \hat{X}_{k,\alpha}) \right) \sigma_K^{-1} n^{-1/2}$$
$$= O_p \left( \tau^p \max_{1 \le k \le K} \left\{ e^{5\|\gamma_k\|_\infty} \|\partial \gamma_k(\theta, s)/\partial s\|_\infty \right\} \sigma_K^{-1} n^{-1/2} \right).$$

From Lemma S3 in [9] and assumption (D1),

$$d_{SW} \left( m_k(\alpha, \Psi, \tau), \hat{m}_k(\alpha, \Psi, \tau) \right) = O_p \left( \tau^p \max_{1 \le k \le K} \left\{ e^{5\|\gamma_k\|_\infty} \|\partial \gamma_k(\theta, s)/\partial s\|_\infty \right\} \sigma_K^{-1} n^{-1/2} \right).$$

$\square$

## J.7   Proof of Theorem 4

**Lemma 5.** *Note that under assumption (S3),*

$$d_\infty(X, X_K) = O_p(1).$$

*Proof of Lemma 5.* Recall that $X_K(\theta, s) = \nu(\theta, s) + \sum_{k=1}^{K} \chi_k \gamma_k(\theta, s)$ and let $Z_K = X - X_K$. For the $k$-th principal component $\chi_k = \int_{\Theta \times S} X(\theta, s) \gamma_k(\theta, s) d\theta ds \le C_2$ from the assumption (T2). We first show that the process $Z_K$ is Lipschitz continuous,

$$|Z_K(\theta_1, s_1) - Z_K(\theta_2, s_2)| \le L_K \left( \|\theta_1 - \theta_2\|_2 + |s_1 - s_2| \right),$$

where $L_k = O(\sum_{k=1}^{K} \lambda_k^{-1})$. From assumption (T2),

$$|X(\theta_1, s_1) - X(\theta_2, s_2)| \le |X(\theta_1, s_1) - X(\theta_2, s_2)| + |X(\theta_1, s_1) - X(\theta_2, s_2)|$$
$$\le C_2 \left( \|\theta_1 - \theta_2\|_2 + |s_1 - s_2| \right).$$

Similarly,

$$|\nu(\theta_1, s_1) - \nu(\theta_2, s_2)| \le C_2 \left( \|\theta_1 - \theta_2\|_2 + |s_1 - s_2| \right),$$
$$|H(\theta_1, s_1, \alpha, t) - H(\theta_2, s_2, \alpha, t)| \le 2C_2 \left( \|\theta_1 - \theta_2\|_2 + |s_1 - s_2| \right).$$

Expanding the eigenfunction $\gamma_k(\theta, s) = \lambda_k^{-1} \int_{\Theta \times S} H(\theta, s, \alpha, t) \gamma_k(\alpha, t) d\alpha dt$,

$$|\gamma_k(\theta_1, s_1) - \gamma_k(\theta_2, s_2)| \leq \lambda_k^{-1} \int_{\Theta \times S} (H(\theta_1, s_1, \alpha, t) - H(\theta_2, s_2, \alpha, t)) \gamma_k(\alpha, t) d\alpha dt$$
$$\leq 2C_2 \lambda_k^{-1} (\|\theta_1 - \theta_2\|_2 + |s_1 - s_2|).$$

Then

$$|X_K(\theta_1, s_1) - X_K(\theta_2, s_2)| \leq |\nu(\theta_1, s_1) - \nu(\theta_2, s_2)| + \sum_{k=1}^{K} |\chi_k| |\gamma_k(\theta_1, s_1) - \gamma_k(\theta_2, s_2)|$$
$$\leq C_2 \left(1 + 2C_2 \sum_{k=1}^{K} \lambda_k^{-1}\right) (\|\theta_1 - \theta_2\|_2 + |s_1 - s_2|),$$

whence

$$|Z_K(\theta_1, s_1) - Z_K(\theta_2, s_2)| \leq C_2 \left(2 + 2C_2 \sum_{k=1}^{K} \lambda_k^{-1}\right) (\|\theta_1 - \theta_2\|_2 + |s_1 - s_2|)$$
$$\leq L_K (\|\theta_1 - \theta_2\|_2 + |s_1 - s_2|),$$

where $L_K = O(\sum_{k=1}^{K} \lambda_k^{-1})$. Similar to Lemma 1 in [35], it follows that

$$\|Z_K\|_\infty \leq 2 \max \left(|\Theta|^{-1/2} |S|^{-1/2} \|Z_K\|_2, L_K^{1/3} \|Z_K\|_2^{2/3}\right). \tag{50}$$

Regarding the $L^2$ norm of $Z_K$,

$$E\left(\|Z_K\|_2^2\right) = \int_{\Theta \times S} E(X - X_K)^2 d\theta ds$$
$$= \int_{\Theta \times S} E\left(\sum_{k>K} \chi_k \gamma_k\right)^2 d\theta ds$$
$$= \sum_{k>K} \lambda_k.$$

Then

$$\|Z_K\|_2^2 = \|X - X_K\|_2^2 = O_p\left(\sum_{k>K} \lambda_k\right). \tag{51}$$

Combining (50) and assumption (S3),

$$d_\infty(X, X_K) = O_p\left(\left(\sum_{k=1}^{K} \lambda_k^{-1} \sum_{k>K} \lambda_k\right)^{1/3}\right) = O_p(1).$$

$\square$

*Proof of Theorem 4.* Writing $X_{iK} = \nu + \sum_{k=1}^{K} \chi_{ik} \gamma_k$ and $\hat{X}_{iK} = \hat{\nu} + \sum_{k=1}^{K} \hat{\chi}_{ik} \hat{\gamma}_k$, note that $\chi_{ik} = \int_{\Theta \times S} X_i(\theta, s) \gamma_k(\theta, s) d\theta ds$ and $\hat{\chi}_{ik} = \int_{\Theta \times S} \check{X}_i(\theta, s) \hat{\gamma}_k(\theta, s) d\theta ds$. By the triangle inequality and Theorem 3,

$$|\chi_{ik} - \hat{\chi}_{ik}| = \left|\int_{\Theta \times S} \left(X_i(\theta, s) \gamma_k(\theta, s) - \check{X}_i(\theta, s) \hat{\gamma}_k(\theta, s)\right) d\theta ds\right|$$
$$\leq \left|\int_{\Theta \times S} X_i(\theta, s) \left(\gamma_k(\theta, s) - \hat{\gamma}_k(\theta, s)\right) d\theta ds\right| + \left|\int_{\Theta \times S} \left(X_i(\theta, s) - \check{X}_i(\theta, s)\right) \hat{\gamma}_k(\theta, s) d\theta ds\right|$$
$$\lesssim d_2(\gamma_k, \hat{\gamma}_k) + d_2(X_i, \check{X}_i)$$
$$= O_p(\sigma_k^{-1} n^{-1/2}).$$

Here the last inequality follows from Proposition 3 and assumption (E1). Assumption (S3) and Lemma 5 imply $d_\infty(\hat{X}_{iK}, X_i) = O_p(1)$. The triangle inequality and results in Theorem 3 and Lemma 4 imply

$$d_\infty\left(X_i, \hat{X}_{iK}\right) \leq d_\infty(\nu, \hat{\nu}) + \sum_{k=1}^{K} d_\infty\left(\chi_{ik}\gamma_k, \hat{\chi}_{ik}\hat{\gamma}_k\right) + d_\infty(X_{iK}, X_i)$$

$$\leq d_\infty(\nu, \hat{\nu}) + \sum_{k=1}^{K} \left(|\chi_{ik} - \hat{\chi}_{ik}| \|\gamma_k\|_\infty + \hat{\chi}_{ik} d_\infty(\gamma_k, \hat{\gamma}_k)\right) + d_\infty\left(X_{iK}, X_i\right)$$

$$= O_p\left(\left(\sum_{k=1}^{K} \sigma_k^{-1} \lambda_k^{-1}\right) n^{-1/2}\right).$$

From (S2) and assumption (T2),

$$\max_{1 \leq i \leq n} \left\{\|X_i\|_\infty, \|\partial X_i(\theta, s)/\partial s\|_\infty, d_\infty\left(X_i, \hat{X}_{iK}\right)\right\} = O_p(1).$$

From assumption (T2) and (T5),

$$R(\tau) = \max_{1 \leq k \leq K} \max_{1 \leq i \leq n} C_4\left(\tau, \|X_i\|_\infty, \|\partial X_i(\theta, s)/\partial s\|_\infty, d_\infty\left(X_i, \hat{X}_{ik}\right)\right)$$

$$= O_p(\tau^p).$$

Regarding the $L^2$ convergence,

$$d_2\left(X_i, \hat{X}_{iK}\right) \leq d_2(\nu, \hat{\nu}) + \sum_{k=1}^{K} d_2\left(\chi_{ik}\gamma_k, \hat{\chi}_{ik}\hat{\gamma}_k\right) + d_2(X_{iK}, X_i)$$

$$\leq d_2(\nu, \hat{\nu}) + \sum_{k=1}^{K} \left(|\chi_{ik} - \hat{\chi}_{ik}| + \hat{\chi}_{ik} d_2(\gamma_k, \hat{\gamma}_k)\right) + d_2\left(X_{iK}, X_i\right)$$

$$= O_p\left(\left(\sum_{k=1}^{K} \sigma_k^{-1}\right) n^{-1/2} + \left(\sum_{k>K} \lambda_k\right)^{1/2}\right).$$

Here we use (51) from Lemma 5 and $d_2(X_{iK}, X_i) = O_p\left(\left(\sum_{k>K} \lambda_k\right)^{1/2}\right)$. Using assumptions (T4) and (T5),

$$d_\infty\left(f_i(\cdot), \hat{f}_i(\cdot, K, \Psi, \tau)\right) = d_\infty\left(\Psi^{-1}(X_i), \Psi_\tau^{-1}\left(\hat{X}_{iK}\right)\right)$$

$$\leq d_\infty\left(\Psi^{-1}(X_i), \Psi_\tau^{-1}(X_i)\right) + d_\infty\left(\Psi_\tau^{-1}(X_i), \Psi_\tau^{-1}(\hat{X}_{iK})\right)$$

$$= O_p\left(C_3(\tau) + R(\tau)\left(\left(\sum_{k=1}^{K} \sigma_k^{-1}\right) n^{-1/2} + \left(\sum_{k>K} \lambda_k\right)^{1/2}\right)\right)$$

$$= O_p\left(\tau^{-(\kappa-p)} + \tau^p \left(\left(\sum_{k=1}^{K} \sigma_k^{-1}\right) n^{-1/2} + \left(\sum_{k>K} \lambda_k\right)^{1/2}\right)\right).$$

From Lemma S3 in [9] and assumption (D1),

$$d_{SW}\left(f_i(\cdot), \hat{f}_i(\cdot, K, \Psi, \tau)\right) = O_p\left(\tau^{-(\kappa-p)} + \tau^p \left(\left(\sum_{k=1}^{K} \sigma_k^{-1}\right) n^{-1/2} + \left(\sum_{k>K} \lambda_k\right)^{1/2}\right)\right).$$

With eigenvalues $\lambda_k = ce^{-\theta k}$ for some constants $c$ and $\theta$, it follows that

$$\sum_{k=1}^{K} \sigma_k^{-1} \sim \sum_{k=1}^{K} e^{\theta k} \sim e^{\theta K}.$$

Similarly,

$$\sum_{k>K} \lambda_k = \sum_{k>K} e^{-\theta k} \sim e^{-\theta K}.$$

Choosing $K(n) = \lfloor \frac{1}{3\theta} \log(n) \rfloor$,

$$d_{SW}\left(f_i(\cdot), \hat{f}_i(\cdot, K, \Psi, \tau)\right) = O_p\left(\tau^{-(\kappa-p)} + \tau^p n^{-1/6}\right),$$

and choosing $\tau \sim n^{1/6k}$,

$$d_{SW}\left(f_i(\cdot), \hat{f}_i(\cdot, K, \Psi, \tau)\right) = O_p\left(\tau^{-(\kappa-p)/6k}\right).$$

$\square$

