# OpenReview forum: "Functional data analysis for multivariate distributions through Wasserstein slicing"
_NeurIPS.cc/2025/Conference — NeurIPS 2025 poster_

### Official Review · Reviewer_qbBC · 2025-06-30

**Clarity:** 3
**Significance:** 2
**Originality:** 2
**Rating:** 4
**Confidence:** 4

**Summary:**

The authors propose an embeddings from the space of multivariate probability distributions into an Hilbert space to enable principal component analysis in functional spaces. While several embeddings of this type exist, the authors emphasize that their proposed embedding is invertible, and therefore allows for the interpretation of modes of variations directly in the space of probability distributions. They also provide the asymptotic convergence of the plug-in empirical estimators of the principal components (i.e. modes of variations), and of the reconstructed density estimators. Their method is illustrated with toy examples and experiments on real-world datasets.

**Questions:**

- Why is it necessary to assume that the support domain $D$ is convex?

- In standard PCA, the explained variance curve typically exhibits an elbow. Did you observe a similar behavior in your experiments?

- I may be missing something, but it is unclear to me whether the embedding proposed in this work is truly original. In what way does it differ from the embedding introduced in [9] for regression purposes?

- Why use the log quantile density transformation instead of the quantile function directly?

- In the Baltimore Longitudinal Study of Aging (BLSA) experiment, one could have approached each point cloud with a Gaussian, and then apply Wasserstein tangent PCA on the estimated covariance instead. How different would have been the results?

- I understand that the family of modes of variation is parametrized by a variable $\alpha$ within a compact interval $[-\alpha_0,\alpha_0]$, which is needed for the proofs. However, what happens as $\alpha_0$ goes to infinity?

- Typos. Line 183 "convergene". Line 503 : Theorem 6 and Corollary 7 do not exist.

**Ethical Concerns:**

["NO or VERY MINOR ethics concerns only"]

**Final Justification:**

The authors propose using an embedding of multivariate a.c. probability measures in a Hilbert space to perform PCA on the measures, by taking advantage of the invertibility of the embedding. Although the use of this invertible mapping is very interesting, I feel that it has not been properly justified by experiments and compared to existing methods. Indeed, it is known that functional PCA will fail when considering probability distributions. In practice, tangent PCA generally gives a "good" approximation, even in "ill-posed cases" (e.g. discrete measures). Thus, in the setting of this paper, namely a.c. distributions supported on a compact convex set, I expect the results to be satisfactory. In my opinion, the paper would strongly benefit from being compared to tangent PCA, as it is unclear if the proposed method is actually "better" than existing methods. Including experiments on the geometry of the proposed embeddded PCA for Gaussian distributions is also important in my view. Subject to these two points, I recommend acceptance of the article.

**Limitations:**

Yes.

**Paper Formatting Concerns:**

No formatting concerns.

**Quality:**

3

**Strengths And Weaknesses:**

The main advantage on using the proposed embedding for principal component analysis of probability distributions lies in its invertibility, which is particularly valuable for representing the modes of variation of PCA in the space of probability distributions. Additionally, the rate of convergence of the estimated truncated density is particularly relevant for assessing a reconstruction error of the method.

My major concern with this paper lies in its comparison to the existing literature. First, in my opinion, the comparison setting is not fair. Indeed, it is well known that performing PCA on densities without accounting for the density constraints leads to poor results. A more relevant baseline would be the classical tangent PCA approach, which consists in mapping the probability measures to the tangent space at the Wasserstein Fréchet mean denoted by $\bar{f}$ (the reference distribution) using the Monge maps (i.e., optimal transport maps from $\bar{f}$ to $f_i$), and then performing functional PCA in the space $L^2(\bar{f})$. This is a standard technique in the analysis of the probability density spade endowed with the Wasserstein distance (see, e.g., [a, b]). Another fair baseline would be to apply a classical kernel mean embedding followed by PCA. Even without the invertibility property, I believe it remains worthwhile to compare the proposed method with these (linear Wasserstein PCA and kernel mean embedding PCA) more robust approaches.

Secondly, a more in-depth analysis of the Gaussian case would have been valuable, especially given the existing literature on this setting. Although the proposed method assumes compact support of the measures, it is still possible to consider truncated Gaussians for comparison. For example, performing PCA directly on the parameters space (mean and covariance) could offer a fair baseline (e.g. with tangent PCA in the Bures Wasserstein space). In the experiments of Setting A, it could have been interesting to visualize the reconstructed $k$-th transformation mode of variation to assess whether the method recovers Gaussian-like behavior. In particular, in my opinion, the importance of embedding invertibility is not adequately emphasized in the experiments.

[a] Wei Wang, Dejan Slepčev, Saurav Basu, John A. Ozolek, and Gustavo K. Rohde. A linear optimal transportation framework for quantifying and visualizing variations in sets of images. International Journal of Computer Vision, 101:254–269, 2013.

[b] Seguy, Vivien, and Marco Cuturi. "Principal geodesic analysis for probability measures under the optimal transport metric." Advances in Neural Information Processing Systems 28 (2015).



------

EDIT : I have increased my rating from 3 to 4 in light of the authors' response to their contributions, which I had certainly underestimated. I also appreciate that the authors plan to add a comparison with tangent PCA and extend the experiment with Gaussian distributions so that readers can better understand the geometry  of the proposed embedding.

---

> ### Author Rebuttal · Authors · 2025-07-30
>
> ## **Response to Reviewer qbBC**
> We thank the reviewer for the insightful comments. We address each concern below:
>
> ---
>
> ### **Fairness of Baseline Comparisons**
>
> We agree that accounting for density constraints is crucial in distributional PCA and appreciate the suggestion to include Tangent PCA and kernel mean embedding PCA. Our current comparison focused on invertible methods for which reconstruction and visualization of modes of variation are well defined.
>
> In contrast, Tangent PCA and kernel mean embedding PCA are known to be non-invertible, and Tangent PCA with general Wasserstein transports is computationally very expensive in dimensions \\( p > 1 \\). It is also heavily dependent on the choice of tuning parameters for regularization (e.g., entropic regularization), where the regularized version can introduce distortion. For example, the modes of variation for large factors may easily venture into the non-invertible area of the linearized space, making invertibility a serious issue.
>
> Due to time constraints in the rebuttal period, we were unable to integrate these additional methods. However, we will include them in the revised version to the extent that well-developed code is available. We will also expand our discussion of these alternative approaches and add references [a] and [b].
>
> ---
>
> ### **On the Convexity Assumption for the Support Domain**
>
> Convexity of the support domain \\( \mathcal{D} \\) is not strictly necessary for the Radon transform itself, which remains well-defined for general multivariate distributions. However, convexity is required due to our use of univariate linearizing (and invertible) transformations for each slice—such as the log quantile density transform (LQD)—which require the density to be positive on the entire support.
>
> This condition is mainly relevant for the theoretical guarantees. In practical implementations, regularization (as discussed in our response to Reviewer kYBw) can be used to address such challenges.
>
> ---
>
> ### **Explained Variance and Elbow Behavior**
>
> We do observe an “elbow” in the cumulative variance explained curve, similar to standard PCA. This pattern is evident in both our synthetic and real-world experiments. For instance, in the blood pressure dataset, the first two components explain 87% of the total variance, after which the contribution of additional components diminishes. We will make this observation more explicit in the revised version.
>
> ---
>
> ### **Originality of the Embedding**
>
> While our embedding builds on the slicing approach from [9], there are important distinctions.
>
> - Our framework generalizes to multivariate distributions using Radon transforms combined with univariate LQD, while [9] focused on univariate regression.
> - Our goal is to develop interpretable modes of variation and convergence guarantees for FPCA in this transformed Hilbert space, supported by theoretical analysis.
>
> The embedding, theoretical development, and application scope in our work are thus quite distinct from those in [9].
>
> ---
>
> ### **Why Use Log Quantile Density (LQD) Instead of the Quantile Function Directly?**
>
> The set of quantile functions is not closed under algebraic operations such as addition or subtraction, whereas log quantile density functions (LQD) are. These operations are crucial for our framework to define principal components and variation modes; see [33] for details.
>
> ---
>
> ### **Alternative Approach for BLSA Using Gaussian Approximation**
>
> We appreciate this suggestion. However, we show that the blood pressure data are not Gaussian, as expected. See our response to Reviewer X7xr for statistical evidence supporting this.
>
> ---
>
> ### **Interpretation as the Parameter \\( \alpha \to \infty \\)**
>
> In our implementation, the modes of variation are defined over a compact interval, typically \\( \alpha \in [-2, 2] \\)(or another compact interval), to ensure both interpretability and boundedness.
>
> While pointwise convergence holds for any \\( \alpha \\), uniform convergence can only be established over compact sets—specifically intervals \\( [-\alpha_0, \alpha_0] \\) for arbitrary \\( \alpha_0 > 0 \\). This is consistent with standard practices in functional data analysis.
>
> ---
>
> ### **Typos and Reference Errors**
>
> Thank you for catching these:
>
> - Line 183: Corrected “convergene” to “convergence”.
> - Line 503: Removed erroneous references to “Theorem 6” and “Corollary 7”.
>
> ---
>
> ### **References**
>
> [a] Wei Wang, Dejan Slepčev, Saurav Basu, John A. Ozolek, and Gustavo K. Rohde.
> A linear optimal transportation framework for quantifying and visualizing variations in sets of images.
> *International Journal of Computer Vision*, 101:254–269, 2013.
>
> [b] Seguy, Vivien, and Marco Cuturi.
> Principal geodesic analysis for probability measures under the optimal transport metric.
> *Advances in Neural Information Processing Systems*, 28, 2015.

---

> ### Comment · Reviewer_qbBC · 2025-08-04
> **Response**
>
> While I acknowledge the contributions of this paper and am happy to increase my score accordingly, I continue to believe that my main concerns have not been addressed, namely :
> - even though tangent PCA can lead to non-invertible principal components, an exponential map can still be used in experiments to recover distributions in the Wasserstein (especially when using discretized distributions). Therefore, in my opinion, a comparison with tangent PCA, widely considered as the baseline method when dealing with curved space such as the Wasserstein space, is essential.
> - an experiment on Gaussian distributions, in the setting A, to plot for example the first two components, would have been very informative in evaluating the quality of the proposed method. In particular, as the authors emphasize that the main contribution is to provide invertible principal components, it would be valuable to include an experimental assessment of whether these components are indeed "good".
> - it should be clarified/emphasized in the paper that the invertible embedding was introduced in [9].

---

> > ### Author Response · Authors · 2025-08-05
> > **Follow-Up Response**
> >
> > ## **Response to Reviewer qbBC – Follow-Up Comment**
> >
> > We sincerely thank the reviewer for the thoughtful follow-up and for considering an increase in the score. We are grateful for the opportunity to improve our work. Below we address the remaining concerns:
> >
> > ### 1. Tangent PCA as Baseline
> > We agree with the reviewer that Tangent PCA is a widely accepted baseline in the literature. Although we originally focused on invertible methods for reconstruction and visualization, we now recognize the importance of including Tangent PCA for completeness. In the revised version, we will incorporate it as a baseline method and discuss both its strengths and limitations (e.g., non-invertibility outside the exponential map domain).
> >
> > ### 2. Gaussian Distribution Experiment
> > We also agree that visualizing the first two principal components under a Gaussian setting is valuable. In the revised simulation Setting A, we will include a new experiment on Gaussian distributions, visualizing both standard PCA and our RLQD approach. This will help assess the interpretability and quality of the learned modes in a well-understood setting.
> >
> > ### 3. Clarification on Invertible Embedding
> > We will clarify and emphasize in the revised version that the invertible embedding was originally introduced in [9].

---

> > > ### Comment · Reviewer_qbBC · 2025-08-05
> > > **Final comment**
> > >
> > > Thank you for considering these suggestions in the revised version of your work.

---

### Official Review · Reviewer_kYBw · 2025-07-02

**Clarity:** 3
**Significance:** 3
**Originality:** 4
**Rating:** 5
**Confidence:** 5

**Summary:**

This paper addresses the challenge of analyzing samples of multivariate distributions by proposing a novel, invertible transformation that maps them into a Hilbert space using a Wasserstein slicing approach. This enables the application of functional data analysis tools—such as functional PCA, modes of variation, and regression—while preserving the ability to recover the original distributions. Theoretical convergence results are established, and the method is demonstrated using joint systolic and diastolic blood pressure data.

**Questions:**

no questions.

**Ethical Concerns:**

["NO or VERY MINOR ethics concerns only"]

**Limitations:**

yes.

**Paper Formatting Concerns:**

no comments.

**Quality:**

4

**Strengths And Weaknesses:**

This paper extends the univariate log quantile density and log hazard transformation approach to multivariate distributions, enabling the use of functional data analysis techniques—such as FPCA, functional modes of variation, and regression—within a Hilbert space framework. The original densities are recovered via a regularized inverse map, and convergence results are established under regularity conditions. The paper also addresses practical challenges related to estimating multivariate distributions from sampled data.

Here are some comments based on my reading.

First, how to handle density functions with multiple isolated support sets?

Second, another limitation of the Radon transformation is the curse of dimensionality.

---

> ### Author Rebuttal · Authors · 2025-07-30
>
> ## **Response to Reviewer kYBw**
> We thank the reviewer for the insightful comments. Below we provide detailed responses to the main points raised.
>
>
> ### **Handling Densities with Multiple Isolated Support Sets**
>
> While the Radon transform remains well-defined for general multivariate densities—including those with non-connected support sets—certain univariate transformations (e.g., log quantile density transformation or log hazard transformation) require that the densities are positive on the domain. This implies that the corresponding probability measures must also be positive on the entire domain, and thus separated subdomains are excluded for the theoretical guarantees to hold.
>
> However, in practical applications, when dealing with densities that exhibit multiple isolated support sets, we adopt a regularization strategy by modeling a perturbed version of the true density:
> $$
> g = \frac{f + \varepsilon}{1 + \varepsilon \cdot |\mathcal{D}|},
> $$
> where \\( \varepsilon > 0 \\) is a small constant and \\( \mathcal{D} \\) denotes the domain that covers the full support of \\( f \\). This modification ensures that \\( g \\) is strictly positive and smooth over \\( \mathcal{D} \\), which enables stable transformation and reconstruction. The original density \\( f \\) can then be approximated either by taking the limit \\( \varepsilon \to 0 \\), or by applying the inverse adjustment:
> $$
> f = g + \varepsilon \cdot g \cdot |\mathcal{D}| - \varepsilon.
> $$
> An alternative practical strategy is to model each disconnected component of the density separately, if prior knowledge of the support structure is available. The trade-off, however, is that this approach requires performing separate PCAs for each component, rather than learning a unified set of principal components across the entire distribution.
>
> ---
>
> ### **On the Curse of Dimensionality and Practical Focus**
>
> Indeed, the curse of dimensionality presents a challenge for multivariate distributional data analysis with the Radon transform. Our method is theoretically applicable in arbitrary dimensions \\( p \\), where assumption (E1) clarifies how the number of within-distribution samples must grow with \\( p \\) and \\( n \\).
>
> The convergence rate explicitly reflects the dependence on \\( p \\), and deteriorates quickly as \\( p \\) increases. Our practical focus is on low-dimensional settings (\\( p = 2 \\) or \\( 3 \\)), which are most commonly encountered in real-world applications (e.g., physiological measurements, environmental variables).
>
> Our theoretical framework still provides general convergence guarantees in the multivariate setting. We will include further discussion of this limitation in the revised Limitations section.

---

### Official Review · Reviewer_moDX · 2025-07-03

**Clarity:** 3
**Significance:** 2
**Originality:** 3
**Rating:** 4
**Confidence:** 2

**Summary:**

This paper proposes a new method to analyze multivariate probability distributions using functional analysis. The key idea is to slice a distribution into many directions using Radon transform and then apply log quantile/hazard transformation to each slice. Such resulting transformation is invertible so it could be mapped back to the original space for interpretations. The authors also prove theoretical guarantees on asy. convergence and did experiments on simulation data to justify the effectiveness of the proposed method.

**Questions:**

1.  How can be the method applied to high dimensional space if it is difficult to get a good slice from Radon transform? How many samples of slice do you need to make sure the method is working?

2. Can the author explain more on the implications of assumptions made for convergence? I'm not really familiar with this proof and it could be helpful to explain more get me have a better understanding.

3. The asymptoic convergence is based on inf sample size. How it applies to finite sample size? I think this is important as in practice, you have limited data usually.

4. Line 160 mentioned "data application play an important role". This looks like disconnected from Section 4.2. Can you explain the important role specifically?

5. I'm not familiar with functional approaches on PCA, but I think the experiment comparison on the standard FPCA and the proposed method seems to be not enough. I asked LLM if there are other methods, it gives me "Wasserstein geodesic PCA or tangent space PCA". Can the author explain it to me if it could be possible to extend the experiment sections and if not why the above methods are not available for comparison?

**Ethical Concerns:**

["NO or VERY MINOR ethics concerns only"]

**Final Justification:**

Concerns have been addressed in the rebuttal.

**Limitations:**

see above

**Paper Formatting Concerns:**

n.a.

**Quality:**

3

**Strengths And Weaknesses:**

Strength: The proposed method is novel and it allows for invertible mapping that allows classical approaches like pca, regression to be applied meaningfully. This does address the limit of current approach.

Weakness: One weakness is that the proposed method by authors might suffer from high dimensional space, as Radon transform requires sample from an informative direction to get a meaningful slice. The experiments are also too simple. I also feel that the paper mainly focuses on combining two existing approaches for application, although it provides some theoretical analysis on asymptotic convergence, the depth of theory may not fully meet the expectation and emphasis typically preferred by NeurIPS.

---

> ### Author Rebuttal · Authors · 2025-07-30
>
> ## **Response to Reviewer moDX**
>
> We thank the reviewer for the constructive feedback. Below we address each concern in detail:
>
> ---
>
> ### **Applicability in High-Dimensional Spaces**
>
> Our method is primarily designed for multivariate but *low- to moderate-dimensional* settings (e.g., 2D or 3D distributions), which are common in practical applications such as joint modeling of anthropometric, physiological, or environmental variables. While the Radon transform may face limitations in very high dimensions due to the curse of dimensionality, our convergence analysis (Section 6) explicitly shows how the estimation error scales with the dimension $p$.
>
> ---
>
> ### **New Approach**
>
> Indeed, the method draws on two existing ideas (LQD and sliced Wasserstein) but introduces new tools—specifically, modes of variation and K-projected representations for distributional data. For both, we develop non-trivial theoretical guarantees with convergence rates.
>
> ---
>
> ### **Number of Slices**
>
> We found the method performs well with 50–75 Radon slices per distribution, sampled uniformly over the unit sphere. This balances computational efficiency with numerical stability. Empirically, we observe that beyond this range, the performance gains saturate, while using too few slices can lead to poor approximation and reconstruction quality.
>
> ---
>
> ### **Implications of Convergence Assumptions**
>
> Our convergence theorem involves multiple assumptions, each serving a distinct purpose:
>
> - **(D1)** and **(F1)** regulate the properties of the input distributions:
>   - (D1) controls the domain of the distributions
>   - (F1) imposes smoothness conditions on the underlying density functions.
>
> - **(T1)–(T4)** provide a generalized framework for the class of transformation-based methods considered in our theorem. Our proposed method (RLQD) satisfies all of these assumptions, as shown through a concrete verification. We present the theorem in general form to allow applicability beyond RLQD.
>
> - **(K1)** and **(K2)** are standard regularity conditions on the kernel function used in multivariate kernel density estimation, necessary to establish convergence guarantees.
>
> Two additional assumptions introduced in Section 6 include:
>
> - **(E1)**  $
> \lim_{n \to \infty} {N(n)}/{n^r} \geq c \quad \text{for} \quad r > 1 + {p}/{4}
> $
> This ensures that the number of within-distribution observations \\( N_i \\) grows sufficiently fast relative to the number of distributions \\( n \\) and the domain dimension \\( p \\).
> As shown in Proposition 3 (Appendix F), \\( N_i \\) directly impacts the accuracy of density estimation. (E1) is critical for uniform convergence when estimating multiple distributions jointly (Proposition 4, Appendix F).
>
> - **(S1)**  $\sigma_K^{-1} \lambda_K^{-1} n^{-1/2} = \mathcal{O}(1)$
> The condition governs the estimation stability of higher-order eigenfunctions. It is only required when the number of components \\( K \\) grows with \\( n \\). Estimating higher-order modes is increasingly sensitive due to the decay of eigenvalues \\( \lambda_k \\) and narrowing spacing \\( \sigma_k \\).
> As shown in Theorem 3 (Appendix G), we have the convergence bound:
> $
> d_\infty(\gamma_k, \hat{\gamma}_k) = \mathcal{O}_p\left( \lambda_k^{-1} \sigma_k^{-1} n^{-1/2} \right)
> $.
> (S1) ensures this remains bounded as \\( n \to \infty \\). These assumptions are standard in FDA and are shown to be achievable. For instance, under exponential eigenvalue decay (e.g., \\( \lambda_k = c e^{-\theta k} \\)), these conditions are easily satisfied (Corollary 1 and Theorem 2).
>
> ---
>
> ### **Finite-Sample Interpretation**
>
> Our theoretical results are asymptotic and provide basic guarantees about convergence as the sample size increases. They do not guarantee finite-sample error bounds, which we will acknowledge explicitly in the limitations section.
>
> ---
>
> ### **Line 160 – Clarification on the Role of Data Application**
>
> We will omit this sentence in the revised version to avoid confusion.
>
> ---
>
> ### **Comparison to Wasserstein Geodesic PCA and Tangent Space PCA**
>
> We appreciate the reviewer’s interest in broader method comparisons.
>
> - **Tangent Space PCA:**
> This approach has been investigated for univariate distributions (e.g., Chen 2020), but its extension to multivariate settings is nontrivial due to the lack of a globally invertible mapping from the Wasserstein manifold to its tangent space. In particular, the exponential map is not defined on the entire Hilbert space but only on a convex cone within it. This poses a significant limitation for practical use, as elements lying outside the invertibility cone cannot be mapped back to the Wasserstein space without relying on ad hoc strategies—such as projecting them onto the cone—which can distort the recovered distributions.
>
> - **Wasserstein Geodesic PCA (WGPCA):**
> While theoretically elegant, WGPCA generally lacks a closed-form inverse map and does not guarantee invertibility for general non-Gaussian multivariate distributions (e.g., Vesseron 2025). This limits its utility for reconstructing densities or visualizing interpretable modes of variation. In contrast, our method offers an explicit and invertible transformation to a Hilbert space, enabling both functional analysis and exact mapping back to the distributional domain.
>
> Additionally, our data application in Section 8 falls in a non-Gaussian regime where we do Mardia's test for multivariate normality; see the response to Reviewer X7xr. This reinforces the advantage of RLQD, which is designed to capture non-Gaussian modes of variation. In contrast, methods like Wasserstein geodesic PCA approaches can only achieve invertibility under a Gaussian assumption. For this reason, we believe it would be inappropriate to emphasize comparisons in purely Gaussian simulation settings when the real-world application of interest is demonstrably non-Gaussian. As such, we focus our comparisons on methods that are both practically applicable and invertible.
>
> ---
>
> ### **References**
>
> - Yaqing Chen, Zhenhua Lin, and Hans-Georg Müller. *Wasserstein regression*. Journal of the American Statistical Association, 118:869–882, 2023.
>
> - Nina Vesseron, Elsa Cazelles, Alice Le Brigant, and Thierry Klein. *On the Wasserstein geodesic principal component analysis of probability measures*. arXiv preprint arXiv:2506.04480, 2025.

---

> > ### Comment · Reviewer_moDX · 2025-08-06
> >
> > Thank you for your detailed and thoughtful rebuttal. I appreciate the clarifications and your efforts to address my comments across both theoretical and experimental aspects. I'm happy to increase my score by 1.

---

### Official Review · Reviewer_X7xr · 2025-07-05

**Clarity:** 2
**Significance:** 2
**Originality:** 2
**Rating:** 5
**Confidence:** 3

**Summary:**

The submission presents an invertible map from the space of multivariate probability distributions to a Hilbert space,  a method to determine modes of variation, and a means of computing these empirically from data. The result builds primarily on the univariate  Hilbert space embedding [33] by Petersen and M'"uller, and draws significantly on [9] for the multivariate case.

Section 2.2 reviews the Radon transform. Since the Radon transform is unstable, in eq. 4 a low pass filter is applied, and the resulting function renormalized to be a valid density function (eq. 5).  The review is very similar to that in [9].

Section 3.3 introduces the Hilbert Space embedding, which is the univariate embedding of the  Radon transform. Section 4.1 recalls the sliced Wasserstein distance, as similarly reviewed in [9, Section 3], and relates it to the Hilbert space embedding of Section 3.3.

Section 4.2 recalls from [33] the method of obtaining "modes of variation" of the Hilbert space embeddings.

In practice, it is usual to have access to data, and not closed-form densities. Thus in Section 5, approximate densities are obtained via a kernel density estimate on the data, and the modes of variation are computed on the basis of these estimates.

Section 6 describes convergence of the modes of variation, under the estimation strategies and with the assumptions made in the paper.  I did not verify the claims.

Section 7 describes synthetic experiments against functional PCA directly on kernel density estimates, treating them as elements of L2, and comparing with RLQD. The fraction of variance explained is higher for the new method.

Section 8 shows a 2-D blood pressure distribution experiment, where the first two components are sufficient to reconstruct most of the variation, and reveal the nature of changes in blood pressure distribution with age.

**Questions:**

l. 79: should be l_[z,\theta} ?  It seems this is an error copied over from [9 p. 5].   In any case not sure why notation l_{z,\theta} is actually needed in the neurips submission.

The notation J_1 in eq. 4 is undefined (we are told J in eq. 3 denotes the Fourier transform). I assume this notation is taken from [9 p. 5], i.e. the 1-D Fourier transform?

References [36] and [37] are the same.

**Ethical Concerns:**

["NO or VERY MINOR ethics concerns only"]

**Final Justification:**

Thank you for the reply. I appreciate the additional experimental comparisons with FPCA, which strengthen the claims made. The updated abstract is also now better aligned with the paper.

I agree with reviewer qbBC on the importance of including tangent PCA experiments, and experiments on Gaussian distributions. The authors have committed to conducting these experiments in a final version of their submission.

I will increase my Rating to 5.

**Quality:**

3

**Strengths And Weaknesses:**

Strengths
----

The paper clearly presents the sliced Wasserstein distance, and demonstrates how to use the Hilbert Space distribution representation implicit in this distance in order to discover modes of variation. The procedure is shown to give a small gain in explaining variance on synthetic data, compared with functional PCA on the density functions. On real data experiments, the resulting modes of variation are interpretable.

The main contribution of this submission appears to be primarily the calculation of the modes of variation in Section (4.2) their  corresponding empirical estimates in Section 5, the convergence results for these modes in Section 6, and the experimental evaluation of the proposed method.

The generalization of the modes of variation in [33] to the multivariate representation in [9], and the corresponding empirical estimates, are a sufficient contribution for NeurIPS.

Weaknesses (theory)
-----

The abstract makes stronger claims than the paper warrants. First, the claim that the multivariate transformation is "introduced" in this submission is overly strong in light of the results of [9] (which the authors cite) - see  the paper summary above for details. Second, "functional regression" is claimed as a contribution, however this does not appear in the paper. If it did, then a NeurIPS submission should demonstrate a thorough experimental comparison against relevant ML methods that solve the same problem, such as

Zoltán Szabó, Bharath K. Sriperumbudur, Barnabás Póczos, Arthur Gretton
Learning Theory for Distribution Regression
JMLR 17(152):1−40, 2016

which the authors of [33] are aware of.

The paper suffers from lossy compression of some aspects of the manuscripts [33,9].   The paper also reads like a longer journal paper rearranged for conference submission. This was not always done with sufficient care, for instance with an entire Appendix H Temperature case study never mentioned in the main document.

Weaknesses (experiment)
----

Figure 2: performance for RLQD and FPCA seems very similar for cases A,B and K=1,2. Only for K=3 is there a larger difference. Thus the advantage on synthetic data seems marginal.

Figure 3: the argument is made that RLQD is useful as it provides an interpretable functional PCA decomposition of the densities. However, for the blood pressure dataset, would FPCA provide an equally interpretable visualization? On real data is there any significant difference in variance explained between RLQD and FPCA?  It is notable that this is omitted, which weakens the claims made for the method.

---

> ### Author Rebuttal · Authors · 2025-07-30
>
> ## **Response to Reviewer X7xr**
> We appreciate the reviewer’s constructive feedback and have taken several steps to address the concerns raised.
>
> ### **Theory-Related Concerns**
>
> - **Abstract revision:** We have rewritten the abstract to more accurately reflect our contributions. Specifically, we no longer claim to “introduce” the multivariate transformation, and instead clarify that the novelty lies in combining Wasserstein slicing with log quantile density transforms in a framework suitable for multivariate distributional data analysis.
>
> - **Functional regression claim:** We agree that the original manuscript may have overstated this. Functional regression is not the focus of this paper, and we have now moved this discussion to the limitations section. The relevant literature (e.g., Szabó et al., 2016) is now appropriately cited.
>
> - **Appendix reference:** We have reviewed the manuscript to ensure that all appendices, including Appendix H (Temperature case study), are explicitly referenced in the main text to avoid confusion.
>
> ---
>
> ### **Revised Abstract**
>
> *The modeling of samples of distributions is a major challenge since distributions do not form a vector space. While various approaches exist for univariate distributions, including transformations to a Hilbert space, far less is known about the multivariate case. We utilize a transformation approach to map multivariate distributions to a Hilbert space via a Wasserstein slicing method that is invertible. This enables the use of functional data analysis tools—such as functional principal component analysis and modes of variation—with the ability to map back to interpretable distributions. We also provide convergence guarantees for the Hilbert space representations under a broad class of such transforms. The method is illustrated using joint systolic and diastolic blood pressure data.*
>
> ---
>
> ### **Revised Discussion and Limitations**
>
> This work presents a transformation-based framework for the functional analysis of multivariate distributions. By combining Radon slicing with univariate log quantile density transforms, we map multivariate densities into a Hilbert space where classical FDA tools such as FPCA and modes of variation become applicable. The regularized inverse transform ensures that the structure of the original density space is preserved. While not the focus of this paper, we note that this framework facilitates distributional regression once vectorized representations are obtained. Limitations include dependence on accurate density estimation from finite samples and the need for regularization due to the fact that the transformed objects lie in a subset of $L^2$.
>
> ---
>
> ### **Experimental Concerns**
>
> We thank the reviewer for their constructive comments regarding the experiments. In the updated manuscript, we have expanded the simulation section to include a more comprehensive comparison between RLQD and FPCA across multiple modes.
>
> - **Variance explained across modes:** While RLQD and FPCA perform similarly in lower modes ($K = 1, 2$), the advantage of RLQD becomes more pronounced in higher modes. This is shown in the updated simulations:
>
>   - **Setting 1:**
>     - \( K = 4 \): RLQD median = 91.2%, FPCA = 81.7%
>     - \( K = 5 \): RLQD median = 95.6%, FPCA = 84.3%
>   - **Setting 2:**
>     - \( K = 4 \): RLQD median = 86.6%, FPCA = 75.4%
>     - \( K = 5 \): RLQD median = 91.3%, FPCA = 82.7%
>
> - **Practical implications:** In our experiments, the advantage of RLQD becomes particularly evident beyond the first two modes, where FPCA’s explained variance tends to plateau earlier. This highlights RLQD’s strength in modeling higher-order structure more effectively, allowing it to achieve robust performance with fewer retained components. This improvement stems not only from RLQD's better capture of signal in higher-order modes but also from its intrinsic modeling approach that incorporates the structural constraints of distributional data directly into the analysis. We have included this part in the revised manuscript.
>
>   When higher-order eigenfunctions are used, the estimation becomes increasingly unstable. As shown in Appendix G (Theorem 3), the convergence rate deteriorates for higher-order components. Specifically, for the $k$-th component, the estimation error scales as $d_\infty(\gamma_k, \hat{\gamma}_k) = \mathcal{O}_p(\lambda_k^{-1} \sigma_k^{-1} n^{-1/2})$ and $d_2(\gamma_k, \hat{\gamma}_k) = \sigma_k^{-1} \mathcal{O}_p(n^{-1/2})$.
>
>   Importantly, a similar convergence rate also holds for traditional FPCA, which means that the instability in estimating high-order components is not unique to RLQD but is a general statistical limitation of functional eigenanalysis. Therefore, reducing the number of retained components is not only computationally advantageous but also statistically critical for ensuring the stability and interpretability of the estimated eigenfunctions and the final representation. RLQD's ability to achieve high explained variance with fewer components is thus a practical strength, especially when modeling complex or noisy data.
>
> - **Blood pressure dataset:** We have also applied FPCA to the blood pressure dataset and observed two key advantages of RLQD: (i) higher variance explained in fewer components and (ii) more interpretable and data-consistent modes of variation, both of which align with our findings from synthetic experiments.
>
>   In terms of variance explained, the first two RLQD components account for 73% and 14% of the total variance, respectively, while the first two FPCA components explain 70% and 11%. Under a standard threshold of 85% cumulative variance, RLQD requires only 2 components for effective dimension reduction, whereas FPCA requires at least 3. This reflects RLQD's superior efficiency and parsimony in real-world settings.
>
>   In terms of interpretability, the eigenfunctions and modes of variation produced by FPCA typically resemble Gaussian-shaped perturbations around the mean, which can be restrictive when modeling data with inherent skewness or non-Gaussian features. In contrast, RLQD captures more flexible, non-Gaussian modes that better reflect the underlying structure of real-world distributions.
>
>   For example, the blood pressure data exhibit clear asymmetry, with a noticeable skew toward lower pressure levels. This is statistically confirmed by Mardia’s test for multivariate normality: both skewness (3063.39) and kurtosis (42.75) statistics are highly significant (p-values < 0.001), indicating a strong departure from multivariate normality.
>
>   These findings demonstrate that RLQD not only provides more informative components with fewer dimensions but also produces representations that are more faithful to the structural properties of complex real-world distributions. We have produced an additional figure with the modes of variation derived from FPCA, which show systematically less variation in the direction of diastolic blood pressure (to be included in the next version).
>
> ---
>
> ### **Clarification of Notation and References**
>
> We thank the reviewer for pointing out these issues:
>
> - **Line 79:** The reviewer is correct — the notation should be \( l_{z,\theta} \) rather than \( l_{u,\theta} \). We have corrected this in the revised version.
>
> - **Equation 4:** We acknowledge that the notation \( J_1 \) was undefined. We now explicitly define \( J_1 \) in the revised text.
>
> - **Duplicate references [36] and [37]:** Thank you for catching this. We have removed the duplicate and updated the numbering accordingly.
>
> ### **References**
>
> - Zoltán Szabó, Bharath K. Sriperumbudur, Barnabás Póczos, and Arthur Gretton. *Learning Theory for Distribution Regression*. Journal of Machine Learning Research, 17(152):1–40, 2016.

---

> > ### Comment · Reviewer_X7xr · 2025-08-05
> > **Reply to authors**
> >
> > Thank you for the reply. I appreciate the additional experimental comparisons with FPCA, which strengthen the claims made. The updated abstract is also now better aligned with the paper.
> >
> > I agree with reviewer qbBC on the importance of including tangent PCA experiments, and experiments on Gaussian distributions. The authors have committed to conducting these experiments in a final version of their submission.
> >
> > I will increase my Rating to 5.

---

### Note · Authors · 2025-08-14

We thank the reviewers for their constructive feedback, which has helped us substantially enhance both the clarity and empirical depth of our work. This paper presents a framework for the functional data analysis of multivariate distributions via an explicit, invertible transformation from the multivariate distributional space to a Hilbert space. Built from Wasserstein slicing combined with the log quantile density transform, this mapping preserves the intrinsic geometry of probability measures while enabling direct application of classical FDA tools such as functional PCA and modes of variation. The invertibility of the transformation ensures that functional components correspond to valid, interpretable distributions, and we provide theoretical convergence guarantees for both the Hilbert space representations and their inverses. We illustrate the method’s ability to capture complex, non-Gaussian structure in practical settings.

During the rebuttal stage, we made several substantive additions and clarifications:

- Additional baseline experiments – Expanded comparisons against FPCA on densities for both synthetic and real datasets, showing RLQD’s higher variance explained with fewer components and more interpretable modes. We also added discussion of tangent Wasserstein PCA and kernel mean embedding PCA, noting their limitations in invertibility and reconstruction.
- Gaussian-case analysis – Introduced a visualization of modes of variation under a Gaussian setting to evaluate how well our invertible components recover known structure.
- Theoretical clarity – Expanded discussion of assumptions, clarified finite-sample interpretation, detailed convergence rates, interpreted key parameters, and corrected notation and references.
- Practical considerations – Discussed challenges and solutions for selecting the number of slices, as well as handling cases where the support domain is not convex.

These updates address key reviewer concerns, strengthen empirical validation against relevant baselines, and further highlight the methodological novelty and interpretability benefits of our invertible mapping framework.

---

### Decision · Program_Chairs · 2025-09-17

**Decision:**

Accept (poster)

**Comment:**

This paper presents a Hilbert space embedding approach to uncover modes of variations in distributions of multivariate probability distributions. The considered multivariate probability distributions are assumed to be compactly supported and possessing positive densities, and the embedding relies on applying apply log quantile/hazard transformations to slices resulting from a Radon transform. While other embeddings of this type exist, the authors emphasize that the proposed embedding is invertible. Asymptotic convergence of the plug-in empirical estimators of the principal components (i.e. modes of variations), and of the reconstructed density estimators are provided. Their method is illustrated with toy examples and experiments on real-world datasets. The paper was referred by four reviewers who lean towards acceptance (two weak accept and two accept gradings). The discussion allowed to improve some points (e.g., X7xr noted before discussion that “the abstract makes stronger claims than the paper warrants” and progresses in that respect were achieved and acknowledged).  Yet qbBC stressed that the paper would strongly benefit from being compared to tangent PCA, as it is unclear if the proposed method is actually "better" than existing methods. Including experiments on the geometry of the proposed embedded PCA for Gaussian distributions is also important in my view. Subject to these two points, I recommend acceptance of the article”, which I concur with.